# *Ap4* is rate limiting for intestinal tumor formation by controlling the homeostasis of intestinal stem cells

Stephanie Jaeckel[1], Markus Kaller[1], Rene Jackstadt[1], Ursula Götz[1], Susanna Müller[2], Sophie Boos[3,4,5], David Horst[2,4,5,6], Peter Jung[3,4,5] & Heiko Hermeking[1,4,5]

The gene encoding the transcription factor TFAP4/AP4 represents a direct target of the c-MYC oncoprotein. Here, we deleted *Ap4* in *Apc^Min* mice, a preclinical model of inherited colorectal cancer. *Ap4* deficiency extends their average survival by 110 days and decreases the formation of intestinal adenomas and tumor-derived organoids. The effects of *Ap4* deletion are presumably due to the reduced number of functional intestinal stem cells (ISCs) amenable to adenoma-initiating mutational events. Deletion of *Ap4* also decreases the number of colonic stem cells and increases the number of Paneth cells. Expression profiling revealed that ISC signatures, as well as the Wnt/β-catenin and Notch signaling pathways are downregulated in *Ap4*-deficient adenomas and intestinal organoids. AP4-associated signatures are conserved between murine adenomas and human colorectal cancer samples. Our results establish Ap4 as rate-limiting mediator of adenoma initiation, as well as regulator of intestinal and colonic stem cell and Paneth cell homeostasis.

[1] Experimental and Molecular Pathology, Institute of Pathology, Ludwig-Maximilians-Universität München, Thalkirchner Strasse 36, D-80337 Munich, Germany. [2] Institute of Pathology, Ludwig-Maximilians-Universität München, Thalkirchner Strasse 36, D-80337 Munich, Germany. [3] DKTK Research Group, Oncogenic Signaling Pathways of Colorectal and Pancreatic Cancer, Institute of Pathology, Ludwig-Maximilians-Universität München, Thalkirchner Strasse 36, D-80337 Munich, Germany. [4] German Cancer Consortium (DKTK), Partner site Munich, Munich D-80336, Germany. [5] German Cancer Research Center (DKFZ), Heidelberg D-69120, Germany. [6] Institute of Pathology, Charité – Universitätsmedizin Berlin, Charitéplatz 1, D-10117 Berlin, Germany. Correspondence and requests for materials should be addressed to H.H. (email: heiko.hermeking@med.uni-muenchen.de)

The TFAP4/AP4 protein belongs to the class of basic-helix-loop-helix leucine zipper (bHLH-LZ) transcription factors (reviewed in Jung and Hermeking[1]). AP4 exclusively forms homodimers, which bind to the E-box motif CAG/CCTG and thereby either repress or activate the expression of target genes. We previously identified the *AP4* gene as a direct transcriptional target of c-MYC[2]. AP4 is expressed in progenitor/transient-amplifying (TA) cells in human colonic crypts, and in colorectal cancer (CRC) in a pattern similar to c-MYC. The prototypic oncogene c-*MYC* is a direct target of the APC (adenomatous polyposis coli) /Wnt (Wingless/Int-1) pathway[3] and an essential mediator of tumor formation induced by inactivation of *Apc* in the intestine[4,5]. Previous studies performed in CRC cell lines or mouse embryonic fibroblasts suggested that AP4 may contribute to the progression of CRC by regulating genes involved in epithelial–mesenchymal transition (EMT) and proliferation[6–8]. However, the organismal function of Ap4 in the intestinal epithelium and its relevance for intestinal tumor formation has so far not been studied using a genetic approach.

The present study shows that inactivation of *Ap4* by deletion leads to decreased adenoma formation in *Apc^Min* mice, which represent a preclinical model of familial *adenomatous polyposis* (FAP)[9,10]. mRNA profiling revealed downregulation of a large number of genes involved in Wnt/β-catenin and/or Notch signaling in *Ap4*-deficient adenomas of *Apc^Min* mice and organoids derived from the epithelium of the small intestine. In line with these regulations, *Ap4*-deficient intestinal organoids and tumoroids show impaired re-growth capacities and therefore decreased stemness. The reduced number of tumors observed in *Ap4*-deficient mice is presumably due to a decrease in the number of functional intestinal stem cells (ISCs). In addition, *Ap4* inactivation causes an increase in the number of Paneth cells. Our results establish Ap4 as a regulator of ISC and Paneth cell homeostasis and as a rate-limiting mediator of intestinal tumor initiation.

## Results

**Role of *Ap4* in intestinal adenoma formation**. Here we determined the effect of *Ap4* deficiency on adenoma formation in the intestine of *Apc^Min* mice, which harbor an inactivating mutation in one *Apc* allele. Upon spontaneous loss of the second *Apc* allele, these mice develop ~50–100 adenomas in the small intestine by the age of 4–6 months. As expected, adenomas in *Apc^Min/+*/*Ap^-/-* mice did not display Ap4 expression, whereas adenomas of *Apc^Min* mice showed elevated expression of Ap4 (Fig. 1a). Approximately 50% of *Apc^Min* mice succumbed to intestinal adenomas by ~180 days of age (Fig. 1b), which was in line with previous reports[11,12]. However, in *Ap4*-deficient *Apc^Min* mice intestinal cancer-related death was delayed on average by 110 days, with heterozygous mice showing an intermediate delay. *Ap4* deficiency was associated with a ~4-fold decrease in the number of adenomas in the small intestines of moribund *Apc^Min* mice, while the size of adenomas increased (Fig. 1c–e). Unexpectedly, the proliferation rate within small intestinal adenomas of moribund *Apc^Min* mice was not affected by loss of *Ap4* (Supplementary Fig. 1). When adenomas of age-matched, 120 days old *Apc^Min* mice were compared, the *Ap4*-deficient mice showed a ~5-fold decrease in the number of adenomas, whereas the size of the adenomas was not affected (Fig. 2a-c). A decreased number of tumors was also detected in the colon of *Ap4*-deficient *Apc^Min* mice when compared with *Ap4*-wild-type *Apc^Min* mice (Supplementary Fig. 2a). However, due to the low incidence of adenomas in the colon of *Apc^Min* mice these differences did not reach statistical significance. The uniform tumor size and the unchanged proliferation rate of tumors in the small intestine (Supplementary

Fig. 2b) in 120 days old mice suggested that the increase in adenoma size seen in moribund animals was most likely due to the increased life-span of *Ap4*-deficient *Apc^Min* mice. The effects of *Ap4* loss on tumorigenesis observed in *Apc^Min* mice were independent of the gender (Supplementary Fig. 2c-e). When we analyzed *Apc^Min* mice with intestinal epithelial cell (IEC)-specific deletion of *Ap4*, which was achieved by crossing *Villin-Cre* with *Ap4^fl/fl* mice, we obtained similar results as for *Apc^Min* mice with germ-line deletion of *Ap4*: that is, in *Ap4^ΔIEC*/*Apc^Min* mice intestinal cancer-related death was significantly delayed by 110 days, with heterozygous mice showing an intermediate delay (Supplementary Fig. 2f). *Ap4^ΔIEC*/*Apc^Min* mice showed a sixfold decrease in the number of adenomas in the small intestines of moribund and a fivefold decrease in the small intestine of 120 days old *Apc^Min* mice, whereas the size of adenomas increases in moribund mice and the size of the adenomas was not affected in 120 days old mice (Supplementary Fig. 2g, h). Epithelial-specific deletion of *Ap4* in *Apc^Min* mice also resulted in a decreased number of adenomas in the colon, although this effect was not statistically significant (Supplementary Fig. 2i). Deletion of *Ap4* in epithelial cells did not affect the proliferation rate of established adenomas of *Apc^Min* mice (Supplementary Fig. 2j). Therefore, the effects of *Ap4* deletion in the germ-line on adenoma formation are presumably intestinal epithelial cell autonomous. Taken together, these results show that *Ap4* is rate limiting for adenoma initiation in *Apc^Min* mice. As c-Myc is a required mediator of intestinal tumor formation in Apc-mediated tumorigenesis, the results imply a pivotal role of Ap4 among the many known c-Myc target genes in mediating intestinal tumor formation.

**mRNA expression profiling of *Ap4*-deficient adenomas**. To identify pathways mediating the effects of Ap4, we compared the mRNA expression profiles of intestinal adenomas that formed in *Apc^Min* mice with and without *Ap4* deletion. By applying next-generation sequencing (NGS), we identified 1459 mRNAs that were differentially regulated with a fold change in expression > 1.5 ($p < 0.05$) due to deletion of *Ap4* (Fig. 3a). Out of these, 954 mRNAs were significantly downregulated, and 505 mRNAs were upregulated in *Ap4*-deficient *Apc^Min* mice (Fig. 2b, c). Notably, pathway analysis showed that mRNAs encoding for proteins involved in EMT, as well as cell cycle regulatory proteins (e.g., E2F targets) were significantly enriched among the down-regulated mRNAs (Supplementary Fig. 3a, Supplementary Data 1). Furthermore, gene set enrichment analysis (GSEA) showed that mRNAs characteristic for Lgr5-positive ISCs[13] were preferentially downregulated in *Ap4*-deficient adenomas (Fig. 4a, b). We validated this finding using additional, previously published ISC-specific gene signatures[14,15] (Supplementary Fig. 3c, Supplementary Data 2). These signatures also showed preferential enrichment among the mRNAs downregulated after deletion of *Ap4*. Moreover, mRNAs encoding for proteins involved in Wnt/β-catenin and Notch signaling, which control the homeostasis of ISCs[16,17], were also preferentially downregulated in *Ap4*-deficient adenomas (Fig. 4a, b, Supplementary Fig. 3c, Supplementary Data 2). Genes downregulated upon deletion of *Ap4* included ISC markers induced by Wnt/β-catenin signaling, such as *Lgr5* and *Ascl2*[18–20], or by Notch signaling, such as *Olfm4*[21], as well as additional direct Wnt/β-catenin and/or Notch target genes with critical functions in the Wnt and Notch signaling pathways, such as *Sox4, Tcf7/Tcf1, Axin2, EphB3, Jag1, Jag2, Hes1* and *c-Myc* (Fig. 4b). Furthermore, *Notch1* itself was downregulated in *Ap4*-deficient adenomas. Taken together, these results imply that Ap4 regulates the homeostasis of ISCs via activating Wnt/β-catenin and/or Notch signaling pathways.

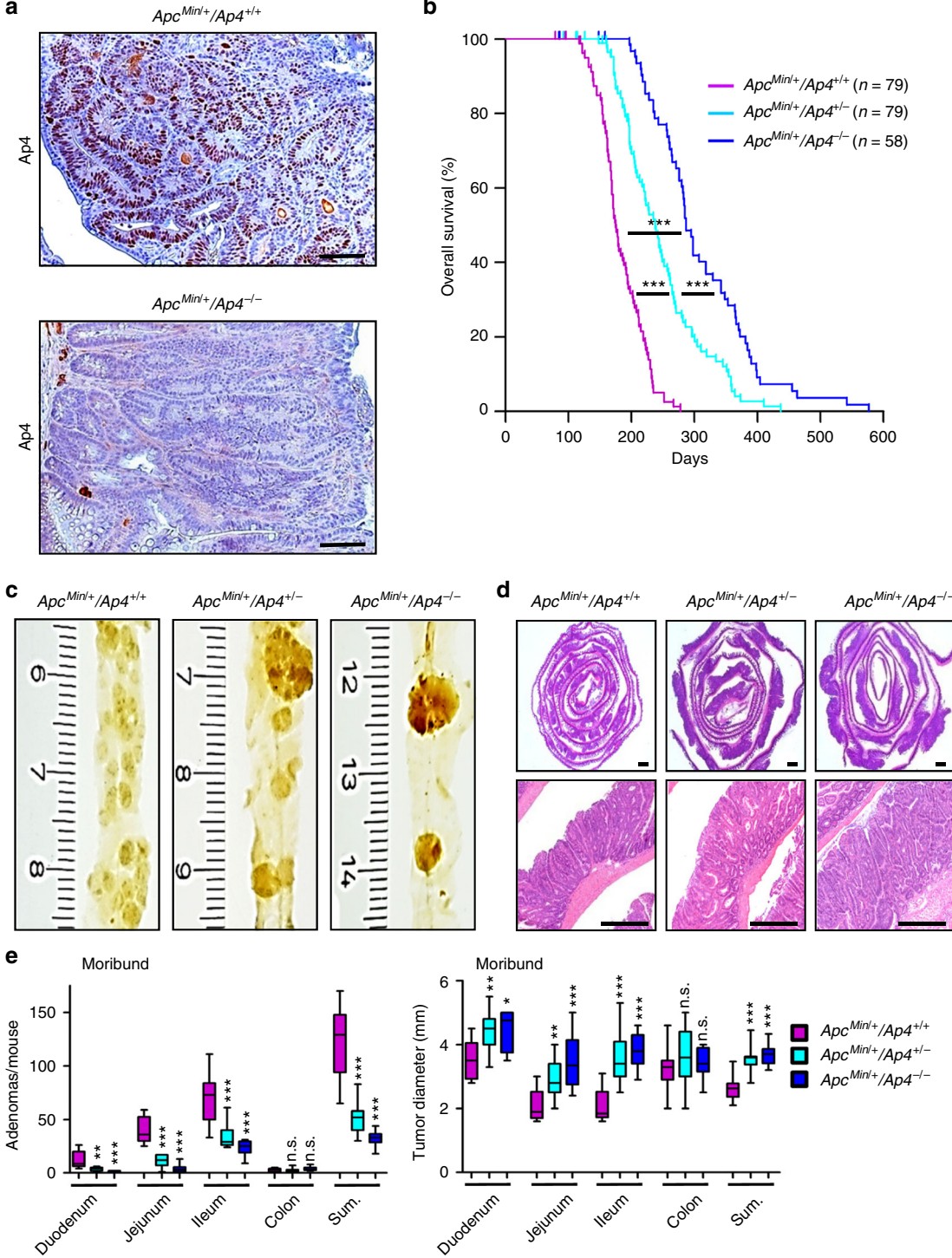

**Fig. 1** Deletion of *Ap4* in *Apc^Min/+* mice prolongs survival and decreases the frequency of adenomas. **a** Immunohistochemical detection of Ap4 in adenomas of moribund *Apc^Min/+* mice with the indicated genotype. Counterstaining with hematoxylin. Scale bar = 50 μm. **b** Kaplan–Meier survival analysis of *Apc^Min/+* mice with the indicated genotypes. Censored mice without intestinal tumor-related death are indicated on the Kaplan–Meier curve as tick marks. **c** Macroscopic pathology of representative polyps in the small intestine (ileum) of moribund *Apc^Min/+* mice with the indicated genotype, scale in cm. **d** Representative sections through rolls of the small intestine stained for hematoxylin and eosin (HE). Scale bar = 500 μm. **e** Quantification of adenoma number/mouse (left panel) and tumor diameter (right panel) in the intestine of six male and six female (*Apc^Min/+*/*Ap4^+/+*), five male and five female (*Apc^Min/+*/*Ap4^+/-*) or four male and four female (*Apc^Min/+*/*Ap4^-/-*) moribund *Apc^Min/+* mice. The box plot extends from the 25th to 75th percentiles. The line in the middle of the box is plotted at the median. The whiskers underneath or above the boxes range from min. to max. value, respectively. **b** Results were subjected to a log-rank test with *p*-values * < 0.05, ** < 0.01, *** < 0.001, n.s. not significant. **e** Results represent the mean ± SD. Results were subjected to an unpaired, two-tailed Student's *t*-test with *p*-values * < 0.05, ** < 0.01, *** < 0.001, n.s. not significant. See also Supplementary Fig. 1

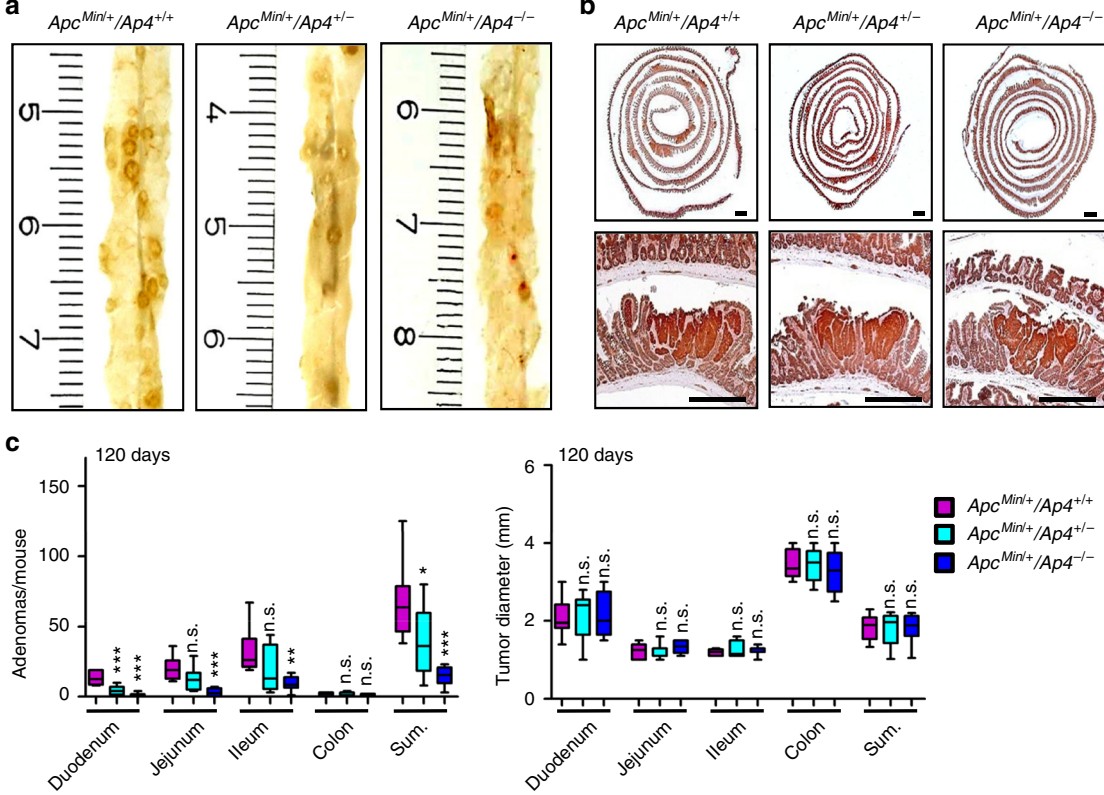

**Fig. 2** *Ap4* deletion decreases frequency but not size of adenomas in age-matched *Apc*^Min/+^ mice. **a** Macroscopic pathology of representative polyps in the small intestine (ileum) of 120 days old *Apc*^Min/+^ mice with the indicated genotype, scale in cm. **b** Representative sections through rolls of the small intestine were stained for β-catenin. Scale bar = 500 μm (upper pictures) or 250 μm (lower pictures). **c** Quantification of adenoma number/mouse (left panel) and tumor diameter (right panel) in the intestine of four male and four female 120 days old *Apc*^Min/+^ mice per genotype. The box extends from the 25th to 75th percentiles. The line in the middle of the box is plotted at the median. The whiskers represent the minimal and maximal values. **c** Results represent the mean ± SD. Results were subjected to an unpaired, two-tailed Student's *t*-test with *p*-values * < 0.05, ** < 0.01, *** < 0.001, n.s.: not significant. See also Supplementary Fig. 2

Recently, Ap4 was shown to maintain a c-Myc-induced transcriptional program in activated T cells[22] and germinal center B cells[23]. In line with these findings, c-Myc target genes were preferentially downregulated in *Ap4*-deficient adenomas (Fig. 4a; Supplementary Data 2). However, the changes in expression of c-Myc target genes observed after deletion of *Ap4* were rather modest compared to the regulations observed in ISC signature or Notch signaling components (Supplementary Fig. 3d). Similarly, E2F targets, though significantly enriched among the downregulated RNAs (Supplementary Fig. 3c), displayed only modest changes in expression that were comparable to those of c-Myc targets (Supplementary Fig. 3d). These modest regulations of c-Myc and E2F targets may explain the lacking influence of *Ap4* deletion on cell proliferation within adenomas.

We exemplarily confirmed the differential regulation detected by NGS using quantitative PCR (qPCR). Thereby, we validated the downregulation of the stem cell markers *Smoc2*, *Lgr5* and *Olfm4*, as well as the repression of several genes involved in the Wnt/β-catenin signaling and/or Notch signaling in *Ap4*-deficient adenomas (Fig. 4c). Consistent with its previously reported repression by AP4[2], *Cdkn1a/p21* was upregulated in *Ap4*-deficient adenomas. Interestingly, we did not detect a change in mRNA or protein levels of *Ctnnb1* (β-catenin) in *Apc*^Min^ adenomas (Fig. 4c, Supplementary Fig. 3e), suggesting that Ap4 directly regulates Wnt/β-catenin target genes.

Next, we analyzed whether Ap4 directly regulates the expression of ISC markers and components of the Wnt/β-catenin

and/or Notch signaling pathways. Our analysis of Ap4 DNA-binding patterns in murine T and B cells[22,23] revealed Ap4 occupancy within the promoter regions of *Ascl2*, *Axin2*, *c-Myc*, *Dll1*, *Dll4*, *EphB3*, *Hes1*, *Hey1*, *Jag1*, *Jag2*, *Notch1*, *Sox4*, and *Tcf7* (Supplementary Fig. 3f). We performed quantitative chromatin-immunoprecipitation (qChIP) analysis to confirm Ap4 occupancy in the murine CRC cell line CT26 at the promoters of the following genes: *Ascl2*, *Dll1*, *Dll4*, *EphB3*, *Hes1*, *Jag1*, *Jag2*, *Notch1*, *Sox4* and *Tcf7* (Fig. 4d). Similar to the promoter of human *CDKN1A/P21*, the murine *Cdkn1a/p21* promoter also contains Ap4-binding sites that showed occupancy by Ap4 (Fig. 4d). Therefore, *Cdkn1a/p21* is a conserved, direct Ap4 target. Taken together, these results suggest that the differential regulation of genes involved in Wnt/β-catenin and/or Notch signaling observed in *Ap4*-deficient *Apc*^Min^ adenomas is a direct consequence of the absence of Ap4 at the respective promoters.

**Analysis of tumor organoids from *Ap4*-deficient *Apc*^Min^ mice.** Our NGS results suggested that Ap4 is involved in maintaining a stem cell-like expression pattern in tumor cells. This was confirmed by in situ hybridization with a probe detecting the mRNA expression of *Lgr5* (Fig. 5a) or *Smoc2* (Supplementary Fig. 4a) in small intestinal adenomas of *Apc*^Min^ mice. Indeed, *Ap4*-deficient adenomas displayed less cells positive for *Lgr5* or *Smoc2* mRNA expression when compared with adenomas expressing *Ap4*. Therefore, the number of tumor stem cells is presumably decreased in the absence of Ap4. Next, we generated tumor

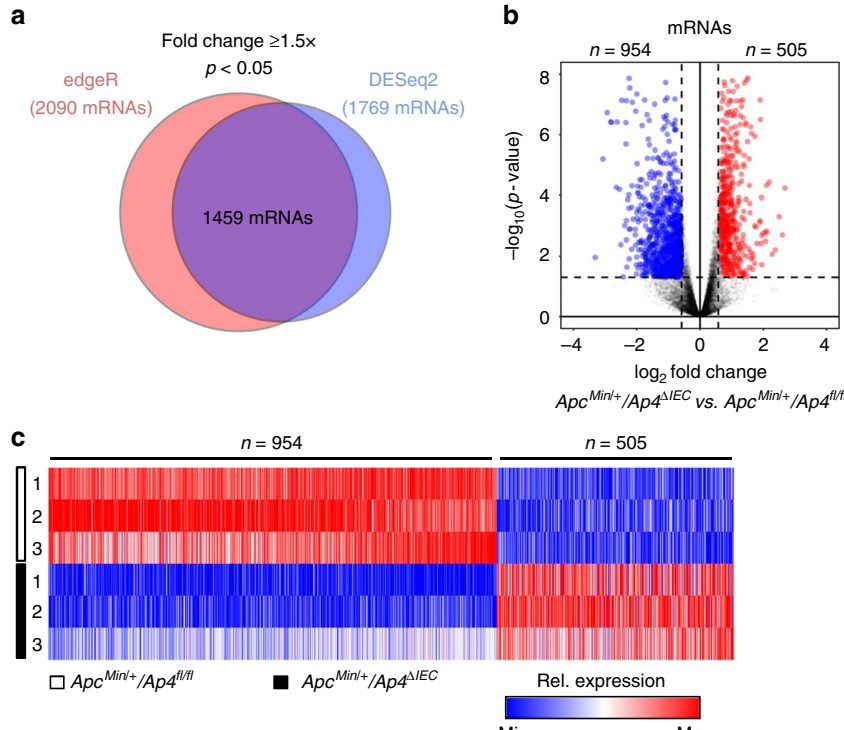

**Fig. 3** Expression analyses of *AP4*-deficient adenomas from *Apc^Min/+* mice. **a** Venn diagram displaying differentially regulated RNAs (fold change ≥ 1.5, $p < 0.05$) in *Ap4^fl/fl* and *Ap4^ΔIEC Apc^Min/+* adenomas as determined by edgeR and DESeq2. **b** Volcano plot and heatmap depicting expression changes between *Apc^Min/+/Ap4^fl/fl* and *Apc^Min/+/Ap4^ΔIEC* tumors from 120 days old mice derived from three female mice (five tumors per mouse) per genotype detected by RNA-Seq. Volcano plot: *p*-values are plotted against the $\log_2$ of the corresponding RNA expression changes in *Ap4^ΔIEC* versus *Ap4^fl/fl* adenomas. Differentially expressed RNAs (*p*-value < 0.05) with a $\log_2$ fold change ≥ 0.58 are indicated in red, with a $\log_2$ fold change ≤ −0.58 are marked in blue. RNAs with 0.58 > $\log_2$ fold change > −0.58 and/or with a *p*-value ≥ 0.05 are represented by gray dots. Dashed vertical lines indicate cut-offs for differential expression. Dashed horizontal line indicates the cut-off for adjusted *p*-values < 0.05 as determined with DESeq2. **c** Heatmap depicting expression changes of differentially expressed mRNAs (fold change ≥ 1.5 and $p < 0.05$ as determined by edgeR and DESeq2) as relative expression levels normalized to the mean expression in the control, *Apc^Min/+/Ap4^fl/fl*, samples for each indicated mRNA. Colors indicate relative expression values from minimum (blue) to maximum (red) for each RNA sample per differentially regulated mRNA. Three biological replicates per genotype were analyzed

organoids using single cells derived from adenomas of *Apc^Min* mice. Indeed, cells directly isolated from *Ap4*-deficient adenomas formed tumor organoids ex vivo with a significantly lower frequency than those derived from *Ap4* wild-type adenomas (Fig. 5b). However, after the first passage ex vivo the *Ap4*-deficient and -proficient tumoroids re-built new tumor organoids with a comparable frequency and growth rate (Fig. 5b). Therefore, Ap4 appears to be required for the initiation, but not for the maintenance of ex vivo cultured intestinal tumoroids. The deletion of *Ap4* in these tumor organoids was accompanied by lower levels of the ISC markers *Smoc2*, *Lgr5* and *Olfm4* when compared with *Ap4* wild-type adenomas (Fig. 5c). Additional genes involved in Wnt/β-catenin signaling and/or Notch signaling, including *Notch1* itself, were also downregulated in *Ap4*-deficient tumor organoids (Fig. 5d). Also at the protein levels, the cleaved, active form of Notch1 (NICD1) and Hes1, which is encoded by a Notch target gene, were decreased in *Ap4*-deficient tumor organoids indicating a decrease in Notch signaling (Fig. 5e). Therefore, the reduced de novo tumor organoid formation capacity may be caused by the downregulation of genes required for in vivo ISC function upon *Ap4* loss.

Subsequently, we isolated small intestinal crypts from *Lgr5*-Cre^ERT2/*Apc^fl/fl* and *Lgr5*-Cre^ERT2/*Apc^fl/fl*/*Ap4^fl/fl* mice. After plating of crypts in Matrigel overlaid with ENR media (containing epidermal growth factor (EGF), Noggin and RSPO1), we acutely deleted *Apc* or *Apc* in combination with *Ap4* in Lgr5-positive stem cells of newly formed intestinal organoids by addition of 4-

hydroxy-tamoxifen (4-OHT). After passaging (passage 1) and seeding the same amount of cells per drop of Matrigel, we switched culture conditions to EN media devoid of RSPO1, in which only Apc-deficient tumoroids can grow. We obtained less de novo formed tumoroids after *Ap4* deletion when compared with Ap4-proficient tumoroids (Fig. 5f, g). This supports the notion that Ap4 has an important role during tumor initiation and confirms the result we previously obtained in vivo. Reassuringly, tumoroids did not form in the absence of RSPO1 and 4-OHT (Supplementary Fig. 4b). During serial passaging, the amount and size of tumoroids was not influenced by the deletion of *Ap4* (Fig. 5f, g, h and Supplementary Fig. 4c). To exclude the possibility that *Ap4*-deficient tumoroids grew due to incomplete deletion of *Ap4*, the complete deletion of *Ap4* was confirmed by genomic PCR (Supplementary Fig. 4d). Taken together, these results confirm a critical role of *Ap4* in the initiation, but not for the maintenance of the tumoroids. These results are in line with the observations initially obtained with *Apc^Min* mice, where *Ap4* loss decreased the number of adenomas but not their size.

**Ap4 regulates the homeostasis of ISCs.** Next, we analyzed the expression of Ap4 and the effect of *Ap4* deletion in the normal, murine intestine. Ap4 protein was detected at the crypt base and in TA cells located above the crypt base in the small intestine (Fig. 6a, left panel). As expected, Ap4 expression was not detectable in the intestinal epithelia of *Ap4* knock-out mice,

indicating that the antibody used here is specific for Ap4 (Fig. 6a, right panel). In mice expressing enhanced green fluorescent protein (eGFP) from an *Lgr5*-promoter, Ap4 expression was detected in eGFP-positive ISCs and TA cells, but not in the adjacent lysozyme-positive Paneth cells (Supplementary Fig. 5a). In *Ap4*-deficient mice, the number of eGFP-positive ISCs was

significantly decreased, indicating that Ap4 is necessary for ISC maintenance (Fig. 6b). Notably, *Ap4*-deficient mice displayed a significant decrease in the number of ISCs positive for *Olfm4* mRNA expression (Fig. 6c). Furthermore, they showed an increased number of Paneth cells in all regions of the small intestine (Fig. 6d): in the ileum each crypt section contained ~8

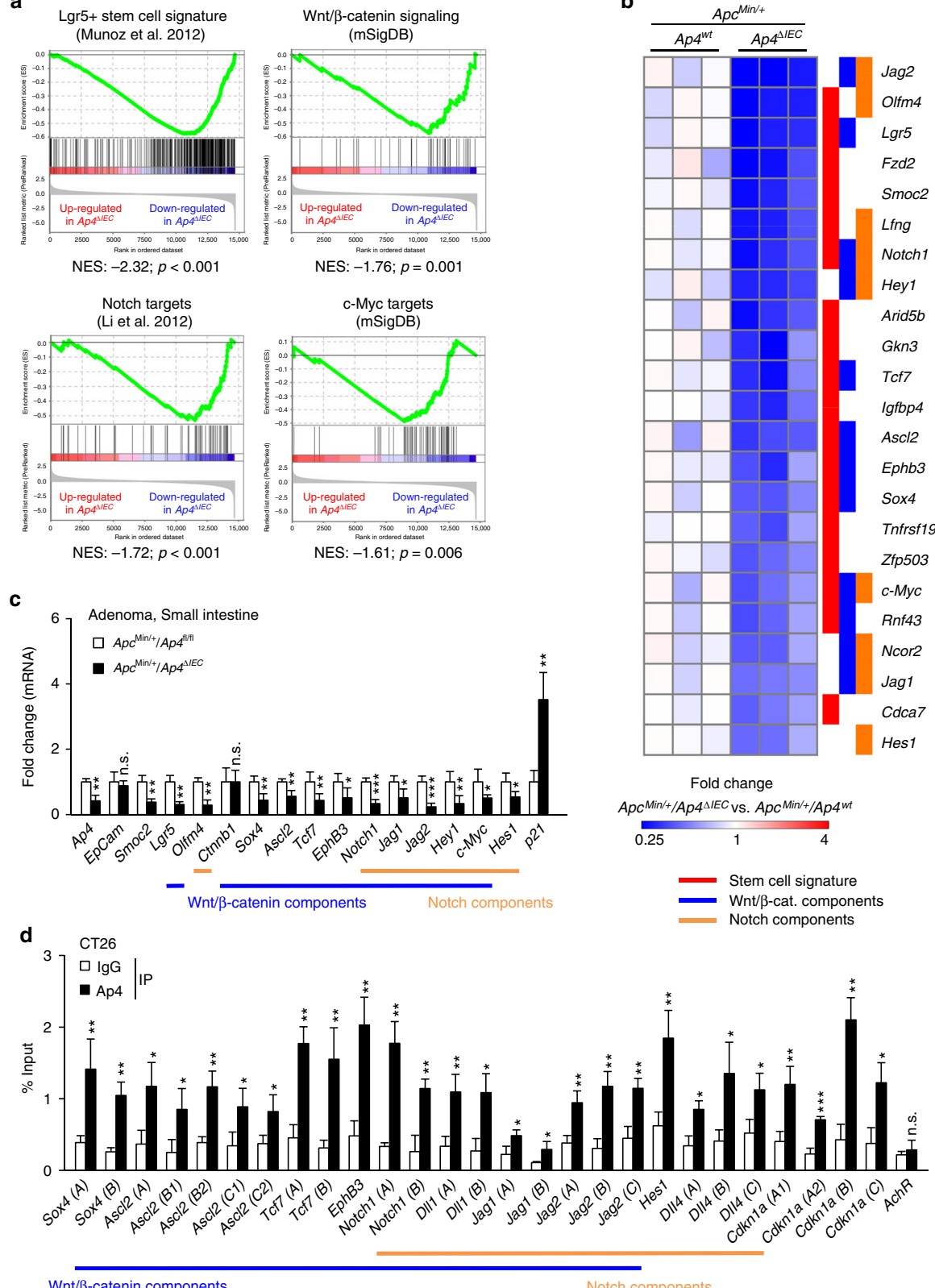

Paneth cells compared with ~5 Paneth cells in wild-type mice. As determined by electron microscopy, Paneth cells of *Ap4*-deficient mice also displayed an increased number of vesicles, which contain antimicrobial proteins, such as lysozyme and cryptdin (Fig. 6e). The length of the small intestine was increased in *Ap4*-deficient mice, presumably as a result of widened crypt bases (Supplementary Fig. 5b). The length of the villi in the ileum was slightly decreased when compared with wild-type mice (Supplementary Fig. 5b). However, the number of TA cells in small intestinal crypts (Supplementary Fig. 5b), the length of the colon and the width of colonic crypts remained unchanged (Supplementary Fig. 5c) The latter presumably due to the absence of classical Paneth cells in the colon. *Ap4* deficiency also resulted in a decreased number of secretory goblet cells in villi of the small intestine (Supplementary Fig. 5d) and in crypts of the small intestine and colon (Supplementary Fig. 5e). Accordingly, mRNA expression of stem cell markers (*Smoc2, Lgr5, Olfm4*) and the goblet cell markers (*Gob5, Muc2*) was significantly decreased, whereas Paneth cell markers (*Lysozyme, Cryptdin*) were significantly increased in the epithelia of the small intestine of *Ap4*-deficient mice (Fig. 6f). We did not detect any effect of *Ap4* deletion on the rate of apoptosis or proliferation in the small intestine (Supplementary Fig. 5f, g). Therefore, these processes did presumably not cause the changes in the numbers of Paneth cells, goblet cells and ISCs observed in *Ap4*-deficient mice. In addition, the effects of *Ap4* deletion described here were independent of the gender of the mice (Supplementary Fig. 5h, i). Furthermore, IEC-specific deletion of *Ap4* had the same effects on the small intestinal and colonic architecture as the germ-line deletion of *Ap4* (Supplementary Fig. 6a-h). Therefore, the effects of *Ap4* loss on ISCs and their derivatives were intestinal epithelial cell autonomous. Interestingly, *Ap4* deficiency also decreased the number of stem cells in the colon (Supplementary Fig. i, j). Age-matched *Apc*^Min mice deficient for *Ap4* also displayed a decreased number of *Lgr5*- and *Smoc2*-positive ISCs per crypt, an increase in Paneth cells, increased length of the small intestine and enlargement of the crypt base of normal intestine, without any change in proliferation or apoptosis in normal epithelium (Supplementary Fig. 7a-g) independent of the gender (Supplementary Fig. 7h). Notably, ISCs have been shown to efficiently form intestinal tumors upon deletion of *Apc*[24] and play a critical role in adenoma and cancer cell self-renewal[25,26]. Taken together, these results suggest that the decreased rate of tumor formation in *Ap4*-deficient *Apc*^Min mice is due to the lower number of functional ISCs in the intestinal crypts.

**Analysis of *Ap4* function in intestinal organoids.** To further analyze the functional relevance of *Ap4* for ISCs, we generated small intestinal organoids by ex vivo culture of small intestinal crypts derived from *Villin-Cre-ERT2/Ap4*^fl/fl mice. After addition of 4-OHT to established organoids, *Ap4* expression was decreased by ~90% within 3 days, which demonstrates efficient, Cre-mediated deletion of the floxed *Ap4* allele in these organoids (Fig. 7a). Upon acute *Ap4* inactivation, organoids showed a

pronounced decrease of ISC markers, an increase of Paneth cell markers, as well as a decrease of goblet cell markers within 3 days after exposure to 4-OHT, whereas organoids derived from *Villin-Cre-ERT2/Ap4* wild-type mice exposed to 4-OHT did not display significant changes in the expression of these markers (Fig. 7a). Importantly, *Ap4*-deficient organoids formed less protrusions (crypt-like structures), when compared with the *Ap4*-expressing organoids (Fig. 7b). As the number of protrusions corresponds to the number of self-renewing ISCs within organoids[27], the decrease in the number of protrusions in *Ap4*-deficient organoids is presumably caused by a decrease in functional ISCs. Taken together, these results suggest that Ap4 is essential for maintaining ISCs in their undifferentiated state and plays an important role in the homeostasis of ISCs and Paneth cells.

**Gene expression profiling of *Ap4*-deficient organoids.** Next, we obtained RNA expression profiles of *Ap4*-deficient and *Ap4* wild-type organoids 7 days after 4-OHT treatment using NGS. Changes in gene expression observed after deletion of *Ap4* were considerably less pronounced in organoids when compared with adenomas. By setting the cut-off for differential expression to a fold change > 1.5 ($p < 0.05$), we identified 693 mRNAs as differentially regulated as a consequence to deletion of *Ap4* (Fig. 7c), with 319 mRNAs being significantly downregulated, and 374 showing upregulation (Fig. 7d). Remarkably, factors involved in Notch signaling and Wnt/β-catenin signaling were significantly over-represented among the downregulated mRNAs (Supplementary Fig. 8a, Supplementary Data 1). In line with the effect of *Ap4* deletion in adenomas, GSEA indicated that mRNAs characteristic for Lgr5-positive ISCs and several factors involved in Wnt/β-catenin and Notch signaling pathways were preferentially downregulated upon deletion of *Ap4* (Fig. 8a, b, Supplementary Fig. 8c, Supplementary Data 2): for example *Sox4, Axin2, EphB3* were downregulated (Fig. 8b, Supplementary Data 2). Downregulated components of the Notch signaling pathway included *Notch1*, and the Notch target gene *Hes1*, as well as the Notch activating ligands *Dll1, Dll3, Dll4* and *Jag2* (Fig. 8b, Supplementary Data 2). Exemplary confirmations of mRNA downregulations of genes important in Wnt/β-catenin signaling and/or Notch signaling upon acute deletion of *Ap4* in intestinal epithelial cell derived organoids are shown in Fig. 8c. The decreased activity of the Notch pathway after Ap4 loss was confirmed by immunohistochemical detection of NICD1 in normal crypts in the small intestine (Supplementary Fig. 8d, e). Not only was the frequency of NICD1-positive cells per crypt lower, but also the intensity of the NICD1 signal was decreased, which indicates a lower activity of the Notch signaling pathway in these cells. Taken together, these results show that Ap4 contributes to the Wnt/β-catenin and Notch transcriptional program in normal intestinal tissue.

Interestingly, the transcription factor Spdef (SAM pointed domain-containing ETS factor) was downregulated in *Ap4*-deficient organoids according to NGS analysis and validated by qPCR (Fig. 8b, c). Spdef regulates the differentiation and

**Fig. 4** *Ap4*-dependent expression profiles in *Apc*^Min/+ adenomas. **a** GSEA comparing gene expression profiles from *Apc*^Min/+/*Ap4*^fl/fl and *Apc*^Min/+/*Ap4*^ΔIEC adenomas from 120 days old mice with Lgr5-positive stem cell signatures[13], Wnt/β-catenin signaling (mSigDB: molecular Signatures Database), Notch target genes or c-Myc target genes (mSigDB). NES: normalized enrichment score, Nom. *p*-value: nominal *p*-value. **b** Heatmap of selected differentially expressed mRNAs (*p*-value < 0.05) from intestinal stem cell gene signatures, Wnt/β-catenin signaling and/or Notch signaling gene signatures analyzed in **a**. The heatmap displays relative fold changes in expression levels normalized to the mean expression in the control, *Apc*^Min/+/*Ap4*^fl/fl, samples for each indicated mRNA. Three biological replicates per genotype were analyzed. **c** qPCR analysis of the indicated *mRNA* derived from tumors from three female mice (five tumors per mouse) per genotype. **d** The murine CRC cells CT26 were subjected to qChIP analysis with Ap4 or IgG-specific antibodies for ChIP. The mouse *acetylcholine receptor* (*AchR*) promoter, which lacks Ap4-binding motifs, served as a negative control. E-boxes used for qChIP analysis are marked in Supplementary Fig. 3. **c, d** Results represent the mean ± SD. Results were subjected to an unpaired, two-tailed Student's *t*-test with *p*-values * < 0.05, ** < 0.01, *** < 0.001, n.s.: not significant. See also Supplementary Fig. 3, Supplementary Data 1 and Supplementary Data 2

maturation of goblet cells[28]. Downregulation of Spdef may therefore contribute to the decreased number of goblet cells observed in *Ap4*-deficient mice.

As observed in adenomas, c-Myc and E2F target genes were significantly enriched among the downregulated RNAs in *AP4*-deficient organoids (Fig. 8a, Supplementary Fig. 8c, Supplementary Data 2) albeit with rather modest fold changes in expression that were considerably less pronounced compared with those of ISC signature and Notch target genes (Supplementary Fig. 8f, Supplementary Data 2). Interestingly, the differential mRNA

expression caused by deletion of *Ap4* was similar in organoids and adenomas as determined by correlation analysis (Fig. 8d). As the organoid derived expression profiles were obtained in the absence of non-epithelial cells or stroma, these findings indicate that the gene expression changes resulting from the inactivation of *Ap4* are largely epithelial cell autonomous. These results suggest that the differential regulation of factors involved in Wnt/β-catenin and/or Notch signaling observed after deletion of *Ap4* in *Apc*^Min mice occurs in normal intestinal epithelial stem cells prior to adenoma development.

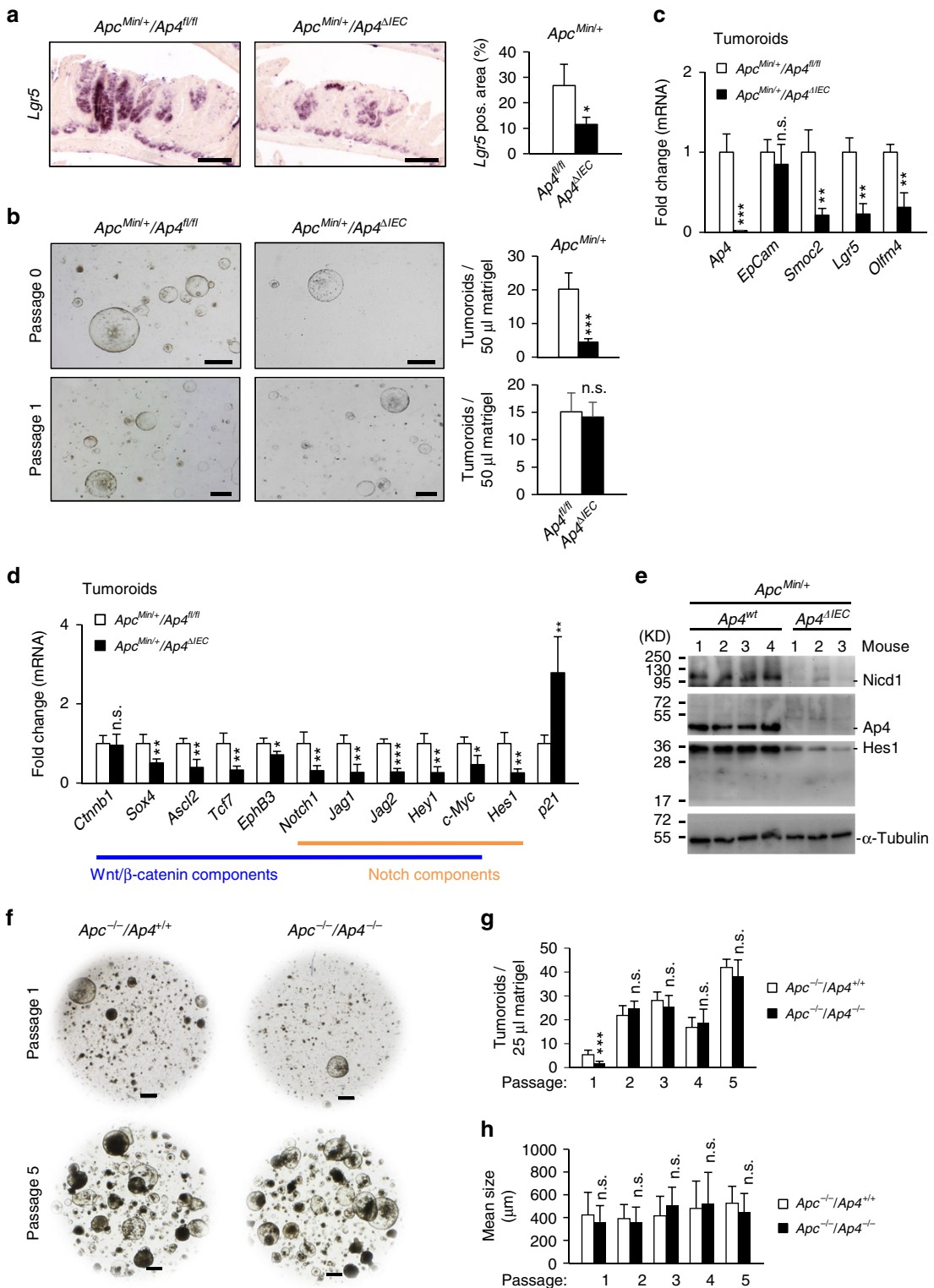

**Regulation of the NOTCH pathway by AP4 in human CRC cells**. In order to determine whether the connection between Ap4 and Notch detected here is conserved between species and relevant to human CRCs, we performed expression and functional analyses in human CRC cell lines. In line with the results described above, the expression of AP4 and NICD1 proteins positively correlated in a panel of five CRC cell lines (Supplementary Fig. 9a). After ectopic AP4 expression in DLD-1 cells, *NOTCH1/NICD1* and the NOTCH-target genes *HES1* and *NRARP* were induced at the protein and mRNA levels (Supplementary Fig. 9b, c). Downregulation of *AP4* by RNA interference decreased NICD1 protein expression in Colo320 cells (Supplementary Fig. 9d). Furthermore, a NOTCH activity reporter plasmid was induced by ectopic AP4 expression, but not by a mutant AP4 protein, which lacks the basic DNA-binding region (Supplementary Fig. 9e). In addition, analysis of public ChIP-Seq data showed open and active chromatin surrounding the sites of AP4 occupancy at the *NOTCH1* promoter, because histone H3K4me1 and H3K27Ac modifications were increased in their vicinity (Supplementary Fig. 9f). When we analyzed ChIP-Seq data, which we had previously obtained after ectopic AP4 expression in the CRC cell line DLD-1[7], we detected AP4 occupancy at the *ASCL2*, *DLL1*, *DLL4*, *EPHB3*, *HES1*, *JAG1*, *JAG2*, *NOTCH1*, *SOX4* and *TCF7* promoters in human DLD-1 CRC cells (Supplementary Fig. 9g). Therefore, AP4 regulates genes involved in WNT/β-catenin and/or NOTCH signaling directly by binding to their promoters in CRC cells.

In addition, inhibition of NOTCH signaling by exposure to the γ-secretase inhibitor Dibenzazepine (DBZ) resulted in a decrease of *AP4* and *c-MYC* expression in SW620 CRC cells (Supplementary Fig. 9h). As expected, the NOTCH-target genes *NRARP* and *HES1* were also repressed. Notably, DBZ suppressed Ap4 protein expression in the small and large intestine of mice (Supplementary Fig. 9i). As expected, inhibition of Notch signaling led to an increase in the number of Paneth cells and goblet cells, as well as to a decrease of Hes1 and c-Myc expression. As we could not obtain experimental evidence for a direct regulation of *AP4* by NICD1 (data not shown), the regulation of *AP4* by the NOTCH pathway is presumably mediated by c-MYC, which represents a known target of the NOTCH pathway[29,30]. Therefore, AP4, the NOTCH pathway and c-MYC form a positive feed-back loop.

**Role of AP4 in human CRCs**. To obtain further evidence for a clinical relevance of AP4 in CRC initiation and progression, we analyzed patient-derived expression data that were generated by the TCGA consortium[31]. Indeed, *AP4* mRNA expression was significantly increased in primary CRCs when compared with normal mucosa in 41 matched normal versus CRC patient samples, as well as in unmatched patient samples representing normal mucosa ($n = 41$) and primary CRCs ($n = 462$) (Fig. 9a). Recently, CRCs were shown to belong to four different molecular subgroups, the so-called consensus molecular subtypes 1–4[32]. In line with the results obtained here, CRCs belonging to the CMS2 subtype, which is characterized by high WNT and c-MYC activity, showed significantly elevated expression of *AP4* when compared with the three other CMS subtypes (Supplementary Fig. 10a). Moreover, in CRCs from the TCGA cohort AP4 expression showed a positive correlation with the expression of mRNAs characteristic for Lgr5-positive ISCs[14], as well as with mRNAs encoding factors involved in Wnt/β-catenin signaling and c-MYC target genes (mSigDB, molecular signature database:[33]; Fig. 9b). Furthermore, *AP4* expression showed a significant positive correlation with *ASCL2*, *TCF7*, *NOTCH1* and *JAG2* expression in 462 primary CRCs in the TCGA cohort (Fig. 9c). As expected, *AP4* expression was also positively associated with *c-MYC* and negatively associated with *CDKN1A/p21* expression.

In order to determine whether the positive correlation between AP4 and NOTCH1/NICD1/HES1 also exists on the level of protein expression in human CRCs, we determined AP4, NICD1 and HES1 expression levels by immunohistochemical analysis of 220 primary CRC samples. For the evaluation of AP4, NICD1 and HES1 expression, we established a four-stage scoring scheme (Supplementary Fig. 10b). AP4 expression was highly concordant with both NICD1 and HES1 expression (Fig. 9d) indicating that the reciprocal regulation between AP4 and the NOTCH pathway also occurs in primary, human CRCs.

## Discussion

This study identified Ap4 as an important, rate-limiting mediator of intestinal adenoma initiation (summarizing model in Fig. 10). As inactivation of *Ap4* led to a decrease in the number of ISCs and an increase in Paneth cells, it is conceivable that the decreased formation of adenomas in the absence of Ap4 is due to the smaller number of bona-fide ISCs that are able to initiate adenomas after acquiring further genetic and epigenetic alterations. This hypothesis is in accordance with the observation that tumor-promoting mutations in ISCs are more efficient in generating tumors than mutations in other, more differentiated intestinal epithelial cells[24,34]. In addition, recent analyses have provided further support for a direct role of stem cell abundance in the determination of tumor frequencies[35]: that is, a strong correlation was found between the tissue-specific stem cell number and the risk to develop a tumor for this particular tissue.

Unexpectedly, deletion of *Ap4* in intestinal epithelial cells had no effect on cellular proliferation. In contrast, acute deletion of

**Fig. 5** Deletion of *Ap4* decreases stemness in adenomas and tumor organoids. **a** Left panel: in situ hybridization of *Lgr5* mRNA. Scale bars represent 100 μm. Right panel: quantification of *Lgr5*-positive area in % in the adenomas from two male and one female mice in at least six adenomas per genotype. **b** Left panel: representative pictures of small intestinal tumor organoids 6 days after isolation (passage 0), upper panel, or 4 days after passaging (passage 1), lower panel. Organoids were isolated from three tumors per mouse from two female and two male mice per genotype. Scale bars represent 500 μm. Right panels: number of tumor organoids per drop of 50 μl Matrigel. A total of 24 drops (passage 0) or 15 drops (passage 1) was analyzed per genotype. **c, d** qPCR analysis of the indicated *mRNA* derived from tumor organoids. **e** Western blot analysis of the indicated proteins. **f** Representative pictures of tumor organoids derived from small intestinal epithelial cells obtained from *Lgr5-CreERT2*$^{+/-}$*/Apc*$^{fl/fl}$ and *Lgr5-CreERT2*$^{+/-}$*/Apc*$^{fl/fl}$*/Ap4*$^{fl/fl}$ mice after treatment with 4-OHT. After isolation, organoids were kept in ENR media (contains EGF, Noggin and RSPO1). Forty-eight hours after isolation, organoids were treated with 4-OHT in a concentration of 100 nM for 48 h to delete *Apc* or *Apc* in addition to *Ap4* in Lgr5-positive intestinal stem cells (ISC). Additional 48 h later, organoids were passaged and EN media (containing EGF and Noggin, but without RSPO1) was used, which selectively allowed Apc-deficient tumoroids to expand (passage 1). Pictures were taken 7 days after passaging in case of passage 1, and 6 days after passaging in case of passage 5. **g** Mean tumor organoid number per drop of 25 μl Matrigel calculated as a mean of a total of 20 drops of 25 μl Matrigel each for passage 1, 3–5 or a total of 11 drops (*Ap4* wt) and 6 drops (*Ap4* ko) for passage 2. **h** Mean tumor organoid size was measured and calculated from Matrigel drops as depicted exemplarily in **d**. **a**, **b**, **c**, **d**, **g**, **h** Results represent the mean ± SD. Results were subjected to an unpaired, two-tailed Student's *t*-test with *p*-values * < 0.05, ** < 0.01, *** < 0.001, n.s.: not significant. See also Supplementary Fig. 4

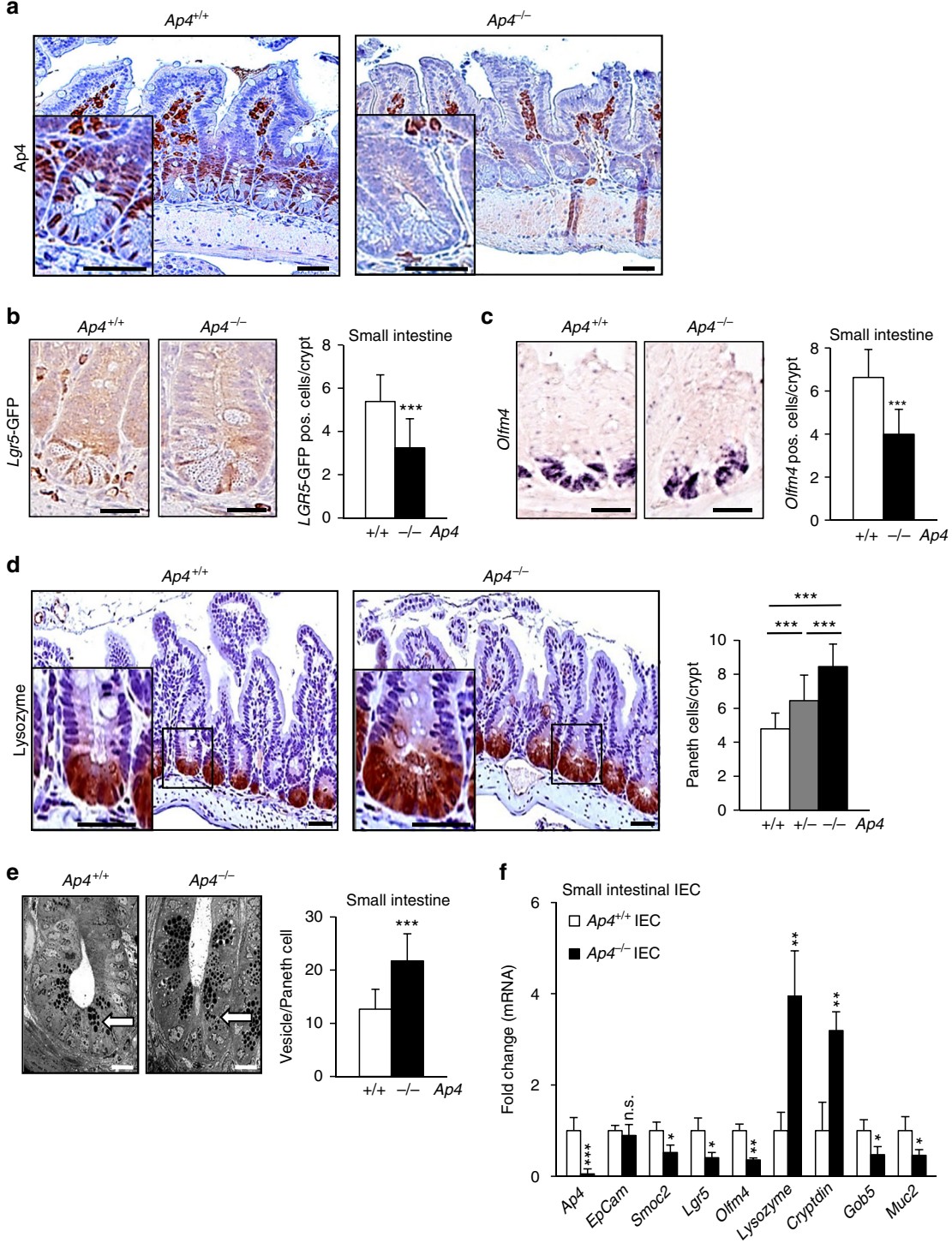

**Fig. 6** Inactivation of *Ap4* causes decrease of ISC and increase Paneth cell numbers. **a** Immunohistochemical detection of Ap4 (brown) in small intestinal tissue, ileum of one male and one female mouse per genotype. Scale bar = 50 μm (25 μm insert), white arrow: site of specific Ap4 expression. Mast cells in the villi display an unspecific staining. Counterstaining with hematoxylin. **b** Left panel: immunohistochemical analyses of Lgr5-eGFP in intestinal sections of 63 days old Lgr5-eGFP mice. Scale bars represent 25 μm. Right panel: quantification of Lgr5-eGFP-positive cells in the crypt base of the ileum from two male and two female mice (130 crypts) per genotype. **c** Left panel: in situ hybridization of Olfm4 mRNA. Scale bars represent 25 μm. Right panel: quantification of Olfm4-positive cells in the crypt base from two male and two female mice (316 crypts) per genotype. **d** Left: immunohistochemical detection of Lysozyme (brown) expressed in Paneth cells. Counterstaining with hematoxylin. Scale bar: 50 μm (25 μm insert). Right: the small intestine/ ileum from two male and two female mice (100 crypts) per genotype was analyzed for Paneth cells/crypt. **e** Electron microscopic analysis of small intestinal crypt base; white arrow: Paneth cells. Scale bar: 25 μm. **f** qPCR analysis of the indicated mRNAs in IECs of the ileum of two male and one female mice per genotype. **b**, **c**, **d**, **e**, **f** Results represent the mean ± SD. Results were subjected to an unpaired, two-tailed Student's *t*-test with *p*-values * < 0.05, ** < 0.01, *** < 0.001, n.s.: not significant. See also Supplementary Figs. 5, 6, 7

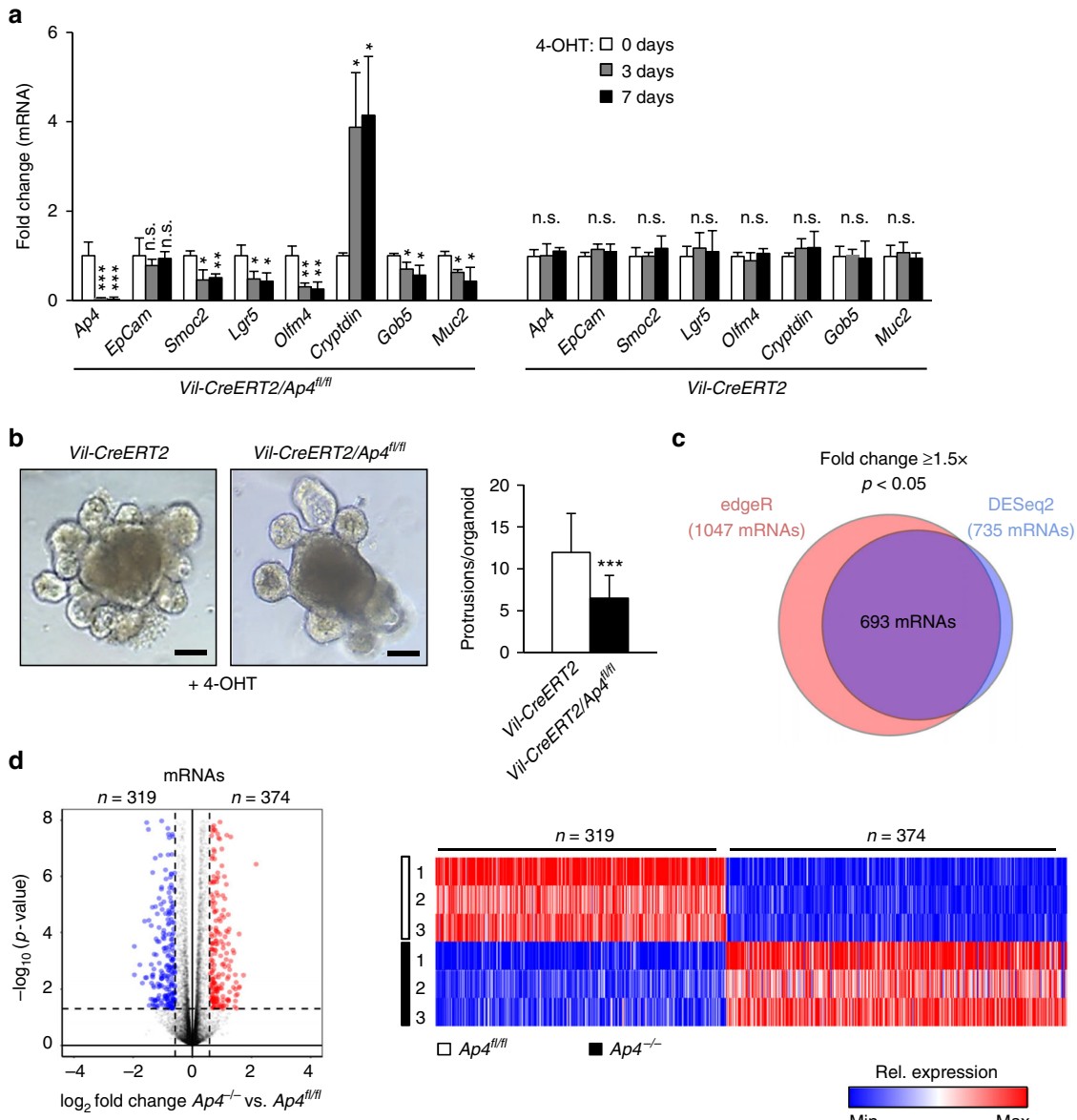

**Fig. 7** Effects of *Ap4* deletion on intestinal organoids. **a** qPCR analysis of the indicated *mRNA* from organoids isolated from three female mice per genotype after passaging and 7 days after Cre activation by 4-OHT. **b** Left panel: representative pictures of small intestinal organoids 7 days after passaging and Cre activation by 4-OHT. Scale bars represent 20 μm. Right panel: quantification of protrusions (crypt-like structures) per organoid derived from three female mice per genotype. A total of 81 organoids were evaluated per genotype. **c** Venn diagram displaying differentially regulated mRNAs (fold change ≥ 1.5, *p* < 0.05) in *Ap4*fl/fl and *Ap4*ΔIEC organoids as determined by edgeR and DESeq2. **d** Volcano plot and heatmap depicting expression changes of differentially expressed mRNAs between *Vil-Cre-ERT2* and *Vil-Cre-ERT2/Ap4*fl/fl organoids, 7 days after Cre activation by 4-OHT, detected by RNA-Seq. Left panel: volcano plot depicting expression changes of differentially expressed mRNAs (fold change ≥ 1.5) from *Ap4*fl/fl and *Ap4*ΔIEC organoids. Downregulated mRNAs are depicted in blue, upregulated mRNAs are depicted in red. RNAs with fold change < 1.5 and/or statistically nonsignificant changes in expression are indicated in black. Dashed vertical lines indicate 1.5-fold change cutoff. Dashed horizontal line indicates the cutoff for adjusted *p*-values < 0.05 as determined with DESeq2. Right panel: Heatmap depicting expression changes of differentially expressed mRNAs (fold change ≥ 1.5 *p* < 0.05 as determined by *edgeR* and *DESeq2*) from *Ap4*fl/fl and *Ap4*ΔIEC organoids. Colors indicate relative expression values from minimum (blue) to maximum (red) for each RNA sample per differentially regulated mRNA. **a**, **b** Results represent the mean ± SD. Results were subjected to an unpaired, two-tailed Student's *t*-test with *p*-values * < 0.05, ** < 0.01, *** < 0.001, n.s.: not significant

*c-Myc* using *Ah-Cre* or *Villin-CreER* alleles in post-natal IECs resulted in defects in proliferation and biosynthetic activity in the small intestine[36,37]. The different result may be due to the different timing of gene inactivation: here we used either germ-line or Villin-Cre-mediated deletion of *Ap4*, whereas the two studies on *c-Myc* employed deletion at least 7 days after birth and later, which was necessary because *c-Myc* is essential during embryogenesis. However, they observed that IEC proliferation was at least in part independent of *c-Myc*. Therefore, IECs may rely on

alternative pathways for promoting cell proliferation. In addition, these differences indicate that Ap4 is responsible for mediating a distinct aspect of c-Myc function and does not simply perform all functions of c-Myc.

The results obtained in tumoroids derived from *Apc*Min mice and in tumoroids generated by acute deletion of *Apc* imply that *Ap4* is important for the initiation but not required for the maintenance of tumoroids. These results are in line with the observations we made in *Apc*Min mice, where *Ap4* loss decreased

the number of adenomas but not their growth/size. Interestingly, *Lgr5* gene expression has been reported previously to be dispensable for ex vivo tumor organoid maintenance[38].

Notably, the deletion of *Ap4* resulted in decreased expression of Notch pathway components including Notch1 itself. Interestingly, Notch pathway inactivation promotes differentiation of ISCs into Paneth cells[21], leading to Paneth cell hyperplasia and a decrease in ISCs[21,39]. However, the number of goblet cells

increases after Notch inhibition, whereas it decreased after *Ap4* inactivation in our study. Therefore, Ap4 loss does not simply recapitulate inhibition of the Notch pathway suggesting that Ap4 regulates additional pathways/genes, which contribute to the differentiation of goblet cells. For example, *Ap4* deficiency resulted in a decrease in expression of the transcription factor Spdef. Interestingly, it was previously shown that ectopic expression of Spdef in the murine intestine promotes the

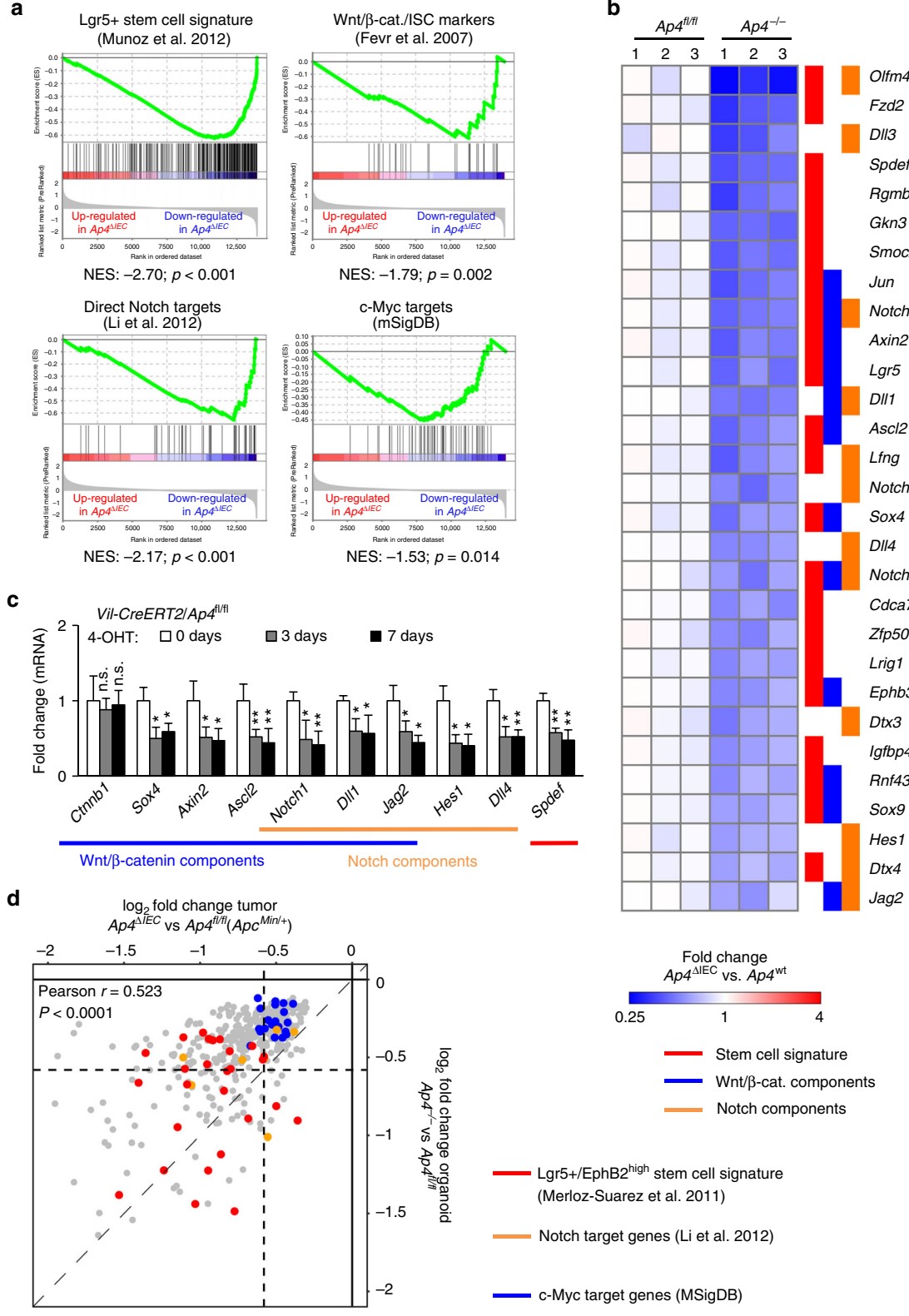

differentiation of goblet cells, whereas it decreased the number of Paneth cells[28]. Accordingly, the decrease of Spdef observed in Ap4-deficient organoids could potentially also contribute to the decreased number of goblet cells detected after Ap4 deletion. Alternatively, the decrease in goblet cells may have been a compensatory response to the increase in Paneth cells caused by Ap4 deletion.

As activation of Notch signaling promotes the initiation of murine intestinal adenomas[40] and inhibition of Notch signaling leads to mitotic arrest and apoptosis in human colon cancer cells[41], the Notch pathway may represent a route via which Ap4 promotes intestinal adenoma initiation. Conversely, Ap4 deletion may prevent adenoma initiation via inhibiting the Notch pathway.

By studying human CRC cell lines, we had previously found that ectopic Ap4 expression is sufficient to activate the Wnt pathway. Our NGS analysis presented here further supports these findings with in vivo evidence, because numerous components of the Wnt/β-catenin signaling pathway were downregulated in Ap4-deficient adenomas and derived organoids. In support of this conjecture, a comprehensive study has recently shown that Ap4 is an important component of Wnt signaling during X. laevis development and acts down-stream of the β-catenin destruction complex to regulate expression of Wnt/β-catenin target genes[42]. Therefore, Ap4 presumably represents an integral component of the Wnt pathway and its activities during development and tumorigenesis. The downregulation of genes involved in Wnt signaling observed here may critically contribute to the decreased number of ISCs observed in Ap4-deficient mice, because the Wnt pathway has been implicated in the maintenance of stemness and suppression of differentiation in ISCs[43–45].

It is well known that a controlled balance of the Notch and Wnt pathway activity is important for the homeostasis of stem cells and cell fate decisions in the intestine and that these two pathways regulate each other at multiple points[46]. Specifically, inhibition of NOTCH1/2 receptors was shown to induce Wnt signaling, which then promoted goblet cell differentiation[46]. The decrease of Wnt pathway component expression after AP4 deletion may therefore alter the expected outcome of a Notch inhibition (i.e., increase of all secretory cell numbers) unto the decrease of goblet cell differentiation observed here.

The increased number of Paneth cells in Ap4-deficient mice might result from an increased propensity of Ap4-deficient ISCs to generate Paneth cell precursors during asymmetric stem cell division. Interestingly, Lgr5 represents an Ap4 target gene[7] and Lgr5 deficiency promotes Paneth cell differentiation in mice[47]. Further discussions of the results and the potential limitations of the Apc^Min mouse model can be found in the Supplementary Discussion.

In conclusion, our study demonstrates an unexpected, central role of Ap4 in ISCs and Paneth cell homeostasis and revealed that Ap4 function is critical for adenoma initiation in a preclinical model of inherited colon cancer. Besides illuminating an important aspect of CRC biology, our results indicate that Ap4 represents a candidate therapeutic target for the treatment of CRCs.

## Methods

**Generation and husbandry of mice**. Targeted ES cells with C57BL/6N background were obtained by homologous recombination with a vector containing the Ap4 exons 2–4 flanked by loxP sites and an intronic neomycin resistance (Neo) cassette flanked by frt sites (scheme in Jackstadt et al.[6]). Ap4^fl/fl mice were generated by injection of targeted ES cells into C57BL/6N blastocyst. The Neo cassette was removed by crossing to flp-mice[48] and germ-line Ap4 knock-out mice were generated by crossing with CMV-Cre+/− mice[49]. Ap4^-/- mice showed no overt phenotype and were born at normal Mendelian ratio. Oligonucleotides used for genotyping are listed in Supplementary Table S1. For analysis of the effect of Ap4 inactivation on the ISC number, we used Lgr5-eGFP-Cre-ERT2+/− mice[50] (obtained from Hans Clevers, University Medical Center Utrecht, The Netherlands) and for specific deletion of Ap4 in intestinal epithelial cells or derived organoids, we used Villin-Cre+/− or Villin-Cre-ERT2+/− mice (obtained from Klaus-Peter Janssen, Technical University Munich, Germany), respectively[51]. APC^Min/+ mice[9,10] were used to analyze the role of Ap4 in intestinal adenoma development (obtained from Marlon Schneider, Ludwig-Maximilians-Universität, München, Germany). Mice were kept in individually ventilated cages with a 12-h light/dark cycle and ad libitum access to water and standard rodent diet. For determination of proliferation rates, 75 mg/kg BrdU (Amersham) in phosphate-buffered saline (PBS) in intraperitoneally injected 1.5 h before mice were sacrificed. All animal experimentations and analyses were approved by the Government of Upper Bavaria, Germany (AZ 55.2-1-54-2532-4-2014).

**Tissue preparation and adenoma counting**. After isolation of intestinal tissue, the colon and small intestine were separated and flushed with PBS to remove stool. The small intestine was dissected into duodenum, jejunum and ileum. The colon and small intestine were opened longitudinally and rolled with the mucosa oriented outwards and fixed in formalin, dehydrated and embedded into paraffin. For evaluation of tumor numbers, each part of the intestine was cut longitudinally and spread on Whatman 3 MM paper. After fixation in formalin, adenomas were counted under a dissection microscope (Zeiss) with 10× magnification.

**Hematoxylin and eosin (HE) and PAS/Alcian blue staining**. Formalin-fixed, paraffin-embedded (FFPE) tissue was cut into 2 μm sections on a rotating microtome (Microm HM355S, Thermo Scientific). The slides were de-paraffinized and stained with hematoxylin (Waldeck) for 6 min followed by eosin (Sigma-Aldrich) for 2.5 min in an automated slide staining device (Tissue-Tek, Prisma). Periodic acid-Schiff (PAS) staining was done by applying Alcian blue pH 1 (Bio Optica) for 10 min followed by periodic acid (Merck) for 5 min, Schiff's reagent (Sigma-Aldrich) for 5 min and counterstaining with hematoxylin (Waldeck).

**Immunohistochemistry**. FFPE tissue was cut into 2 μm sections on a microtome and de-paraffinized. After antigen retrieval, slides were incubated with primary antibody (the primary antibodies used are listed in Supplementary Table 4) for 1 h at room temperature and washed with TRIS-HCL buffer (pH 7.5) followed by a secondary antibody. Antibodies were detected with the ABC kit using DAB (Vector and Dako) for brown stainings or AEC (Thermo Fisher Scientific) for red stainings. The slides were counterstained with hematoxylin (Vector) and mounted with Roti®-Histokitt II (Roth). All stainings were performed with the respective IgG control (Supplementary Table 4) as a negative control and without primary antibody as a system control. Images were captured on an Axioplan2 imaging microscope (Zeiss) equipped with an AxioCamHRc Camera (Zeiss). For analysis of cleaved caspase-3, the AxioVision Software (Zeiss) was used to measure the area for each tumor in mm$^2$.

**Fig. 8** NGS analysis of intestinal organoids after deletion of Ap4. **a** GSEA comparing gene expression profiles from Vil-Cre-ERT2 and Vil-Cre-ERT2/Ap4^fl/fl organoids 7 days after Cre activation by 4-OHT with Lgr5-positive stem cell signatures[13], β-catenin regulated/ISC-specific genes[60], Notch targets genes[61] or c-Myc target genes (mSigDB: molecular Signatures Database). NES: normalized enrichment score, Nom. p-value: nominal p-value. **b** Heatmap depicting expression changes of selected differentially expressed mRNAs (p-value < 0.05) from stem cell gene signatures, Wnt signaling and/or Notch signaling gene signatures analyzed in **a**. The heatmap displays relative expression levels normalized to the mean expression in the control (Vil-CreERT2) samples for each mRNA. Three biological replicates per genotype were analyzed. **c** qPCR analysis of the indicated mRNA of organoids with the indicated genotypes 3 days and 7 days after 4-OHT induction. **d** Scatter plot displaying the correlation of expression changes of the 424 mRNAs significantly (p < 0.05) downregulated in both adenomas and organoids (shown in gray).The mRNAs from stem cell gene signatures, Notch target gene signatures and c-Myc target genes analyzed in Figs. 4a and 8a in both adenomas and organoids are highlighted with the indicated colors. The Pearson correlation coefficient of all 424 mRNAs downregulated in both adenomas and organoids and statistical significance are indicated. **c** Results represent the mean ± SD. Results were subjected to an unpaired, two-tailed Student's t-test with p-values * < 0.05, ** < 0.01, *** < 0.001, n.s.: not significant. See also Supplementary Fig. 9, Supplementary Data 1 and Supplementary Data 2

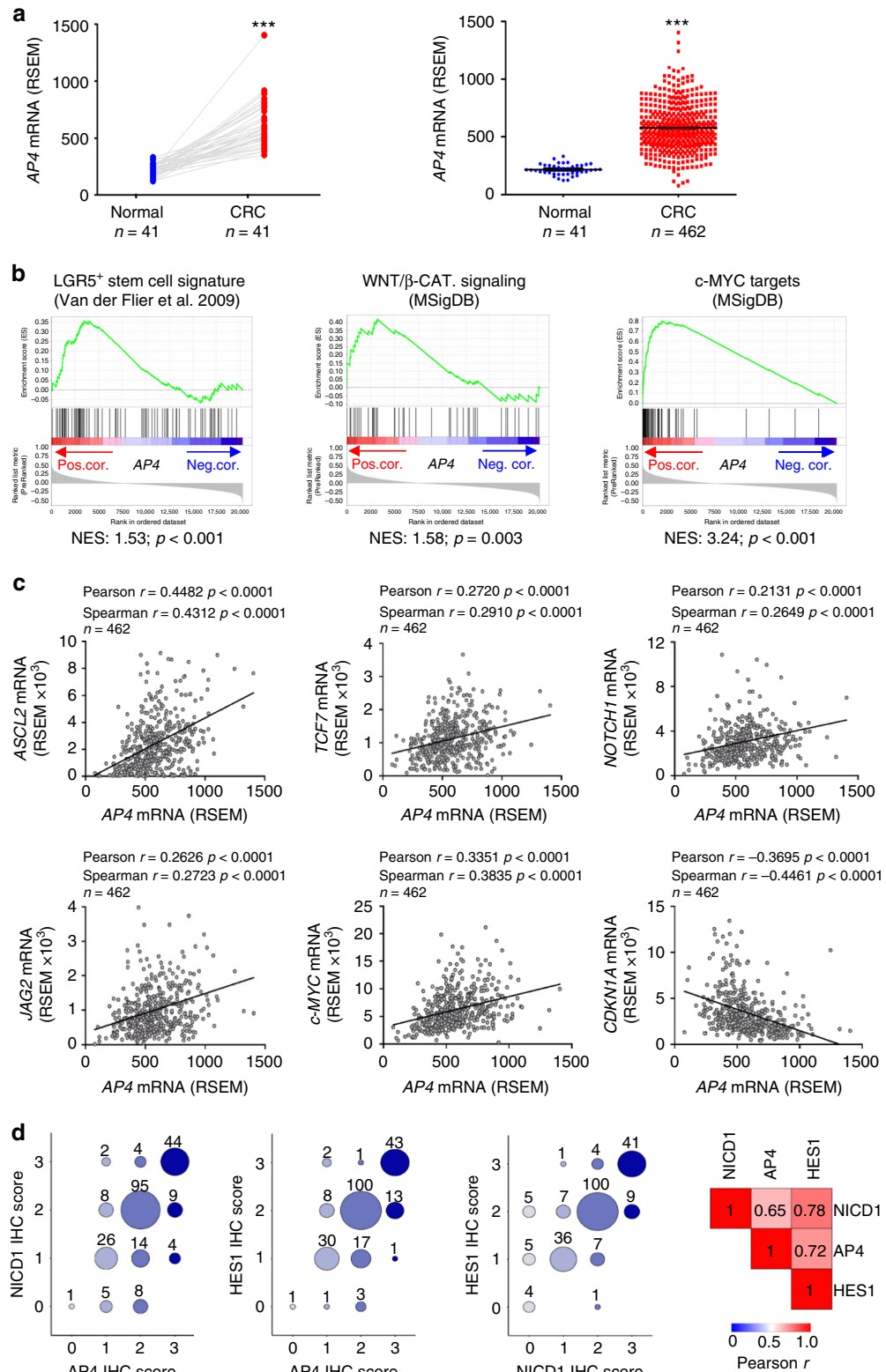

**In situ hybridization**. For detection of ISCs with an *Olfm4* mRNA probe, the Bluescript II plasmid p695-pBS-mOlfm4 (kindly provided by Prof. Hans Clevers) was linearized by using a *Not*I restriction enzyme. The pCMV6 entry plasmids containing the *Lgr5* or *Smoc2* open reading frame (ORF) were obtained from Origene with the catalog number MR219702 or MR207121, respectively. Both ORFs were cloned into the pBSII KS plasmid by using the restriction enzymes *Not*I and *Kpn*I. The pBSII KS-mSmoc2 was linearized with the *Kpn*I restriction enzyme and the pBSII KS-mLgr5 was linearized with the *Bcl*I restriction enzyme. The *Olfm4*, *Lgr5* and *Smoc2* RNA probe has was generated by an in vitro transcription

reaction with a RNA-T7 Polymerase by using the DIG Northern Starter Kit (Roche Diagnostics). During the transcription reaction, the probe was labeled with digoxigenin (DIG). The in situ hybridization was performed on freshly prepared 8 µm paraffin sections as described[52].

**Isolation of IECs**. Each part of the intestine was dissected longitudinally and cut into small pieces. IECs were isolated by shaking the tissue in Hanks' Balanced Salt solution/ethylene-diamine-tetra-acetic acid (HBSS/EDTA) at 37 °C for 10 min. The

**Fig. 9** *Ap4* mRNA expression and associations in human CRCs deposited at TCGA and its correlative analysis of AP4, NICD1 and HES1 protein expression in human CRCs. **a** *Ap4* RNA expression levels from matched patient samples from normal mucosa and tumor tissue (left), or unmatched normal mucosa and tumor tissues (right). RNA expression data were obtained from publically available TCGA colorectal adenocarcinoma (COAD) datasets[31]. Left panel: matched samples, 41 normal mucosa and 41 tumor tissue samples. Right panel: 41 normal mucosa samples and 462 tumor tissue samples were used. **b** Expression of human homologs of the Lgr5-positive stem cell signature[14], members of the HALLMARK Wnt signaling gene set[33] or c-Myc target genes[33] are strongly linked with *Ap4* expression in human tumors. All genes within the TCGA datasets from human colorectal tumors were pre-ranked by expression correlation coefficient (Pearson *r*) with Ap4 in descending order from left (positive correlation) to right (negative correlation) and analyzed by GSEA. **c** Scatter plots displaying pairwise comparisons of mRNA expression levels [RSEM] of the indicated genes for 462 TCGA COAD samples. The correlation coefficients and *p*-values of Pearson´s and Spearman's correlations, as well as the linear regression trend line are indicated. **d** Left: dot plots displaying pairwise comparisons of IHC staining scores (see Supplementary Fig. 10b for more information) for AP4, NICD and HES1 from 220 human CRC tumor microarray (TMA) samples. Dot sizes represent tumor sample numbers. Right: Heatmap displaying Pearson *r* correlation coefficients of IHC scores obtained from 220 tumor samples stained for AP4, NICD1 and HES1 expression. **a** Left panel: results were subjected to a paired, two-tailed Student's *t*-test with *p*-values * < 0.05, ** < 0.01, *** < 0.001. **a** Right panel and **b** results represent the mean ± SD. Results were subjected to an unpaired, two-tailed Student's *t*-test with *p*-values * < 0.05, ** < 0.01, *** < 0.001, n.s.: not significant. See also Supplementary Fig. 10

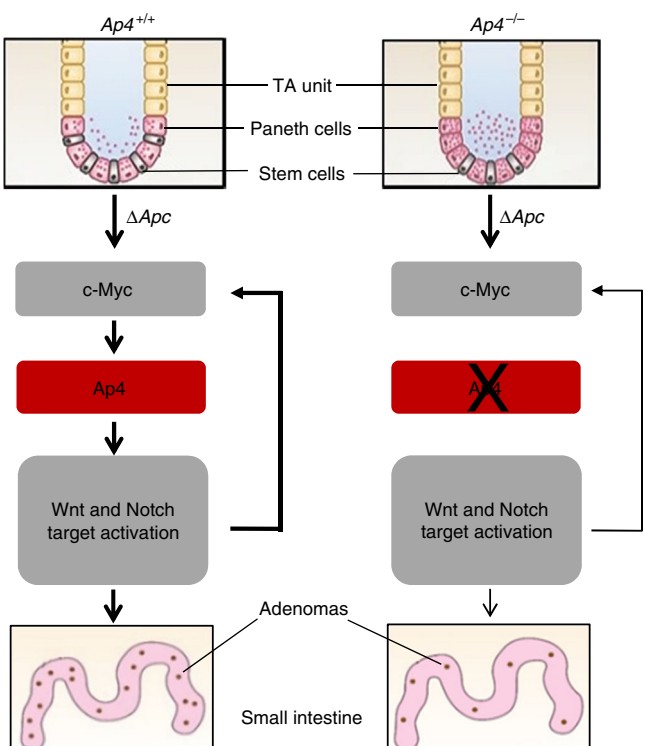

**Fig. 10** Model summarizing the results of this study. The results are explained in the context of the previously described biology of the intestinal crypt: The homeostasis of the intestinal epithelium is regulated by the Wnt/β-catenin and Notch pathways, which control stem cell maintenance and differentiation[16,17]. In the small intestine, the crypt base harbors 4–6 intestinal stem cells (ISCs), which are surrounded by supporting Paneth cells. The Lgr5-positive ISCs divide asymmetrically to self-renew and to generate highly proliferative, Lgr5-negative transit-amplifying (TA) cells, which differentiate into absorptive enterocytes or secretory cells, such as goblet, tuft and entero-endocrine cells[16–18,63]. Paneth cells are derived from Lgr5-positive precursors generated by asymmetric division of ISCs, which proliferate slowly and therefore represent label-retaining cells (LRCs)[64]. In the present study, we found that deletion of *Ap4* decreased the number of ISCs and increased the number of Paneth cells. Furthermore, we found that the Wnt and Notch pathways were downregulated in the absence of Ap4. The decreased number of ISC presumably resulted in the decreased number of adenomas in Ap4-deficient *APC^{Min/+}* mice, since Lgr5-positive ISCs were shown to represent the cells of origin for intestinal cancer[24,34]

supernatant including IECs was centrifuged and the pellet was washed with ice-cold PBS, frozen in liquid nitrogen and stored at −80 °C until RNA isolation.

**Crypt isolation and organoid culture.** Crypt isolation and organoid culture was performed as described before[53]. The small intestine was opened longitudinally and the villi were scraped off under a dissection microscope by using a surgical blade. The intestine was cut into small pieces and incubated in 8 mM EDTA in HBSS for 5 min to remove the rest of the villi followed by an additional incubation in EDTA for 30 min at 4 °C. Isolated crypts were washed in advanced Dulbecco's modified Eagle's medium (DMEM)/F12 (Gibco Life Technologies) containing Glutamax (Gibco Life Technologies) and Hepes (Gibco Life Technologies), passed through a 100 µm cell strainer and either frozen in liquid nitrogen (IEC) or counted and pelleted for culturing. For each well in a 24-well plate, 200 crypts were mixed with 50 µl of growth factor reduced, Phenol red-free Matrigel (Corning). After polymerization of the Matrigel, 600 µl of crypt culture medium was added. Crypt culture medium consists of advanced DMEM/F12 supplemented with 1:100 penicillin/streptomycin (Gibco Life Technologies), 1:100 Hepes, 1:100 Glutamax and the following growth factors: 100 ng/ml Noggin (Preprotech), 1:100 N2 (Gibco Life Technologies), 1:50 B27 retinoic acid free (Gibco Life Technologies), 50 ng/ml EGF (Preprotech), 500 ng/ml RSPO1 (Sinobiological), 10 µM Y-27632 (MedBiochem Express) was added only the first 2 days after isolation/culturing or passaging to avoid anoikis, and 100 ng/ml Wnt-3a (Abcam) was added only the first 2 days after isolation/culturing. Crypt culture medium was changed every 2 days. Once a week, organoids were passaged at a 1:6 ratio. For passaging, organoids were removed from Matrigel and dissociated mechanically into single-crypt domains before they were transferred into fresh Matrigel. 4-OHT (Sigma-Aldrich), diluted in ethanol, was added to the crypt culture medium to a final concentration of 500 nM for at least 12 h. As a control, the same volume of ethanol without 4-OHT was added to the crypt culture medium. RNA was isolated using Trizol (Invitrogen) and the RNeasy Mini Kit (Qiagen).

For generation of tumoroids, intestinal adenoma cells from three tumors for each *Apc^{Min/+}* mouse were isolated by lysis in DMEM containing 4000 units Collagenase Type IV (Merck Millipore) and 125 µg/ml Dispase Type II (Sigma-Aldrich). Single cells were embedded in Matrigel and seeded in 24-well plates (15,000 single cells per 50 µl Matrigel per well). The tumor organoid culture medium was formulated as described before[38]. Crypt culture medium (advanced DMEM/F12 supplemented with 1:100 penicillin/streptomycin, 1:100 Hepes and 1:100 Glutamax and growth factors (1:100 N2, 1:50 B27, 50 ng/ml EGF). In all, 10 µM Y-27632 was added only the first 2 days after isolation/culturing or passaging to avoid anoikis. Counting of the number of organoids per well (six wells per mouse) was performed 6 days after isolation before the first passaging. Passaging was performed as described above for normal organoids. RNA was isolated using Trizol and the RNeasy Mini Kit. For western blot analysis, tumor organoids were lysed in RIPA lysis buffer (50 mM Tris/HCl, pH 8.0, 250 mM NaCl, 1% NP40, 0.5% (w/v) sodium deoxycholate, 0.1% sodium dodecylsulfate, Complete Mini protease inhibitors (Roche Diagnostics)).

For tumoroid formation after acute loss of *Apc* or *Apc* and *Ap4*, crypts were isolated as described above. For each well in a six-well plate, approximately eight drops of 25 µl Matrigel and 50 crypts each were plated and overlaid with ENR media (containing EGF, Noggin and RSPO1) as described above. Forty-eight hours after isolation, organoids were treated with 100 nM 4-OHT in ENR media for 48 h. After passaging an additional 48 h, the culture conditions were switched to EN devoid of RSPO1 to select for tumoroid growth. Pictures of tumoroids were taken with a Nikon AZ-100 macroscope.

**Tissue microarrays and IHC analysis of clinical samples.** Colon cancer specimens from patients that underwent surgical resection at the University of Munich (LMU) were drawn from the archives of the Institute of Pathology. Specimens were anonymized, and the need for consent was waived by the institutional ethics committee of the Medical Faculty of the LMU. Tissue microarray (TMAs) with

samples of 225 (220 evaluable) stage II primary CRC cases were generated with six representative 1 mm cores of each case. In all, 5 µm TMA sections were de-paraffinized and stained with primary antibodies (listed in Supplementary Table 4) on a Ventana Benchmark XT Autostainer with UltraView Universal DAB and alkaline phosphatase detection kits (Ventana Medical Systems). The stainings were evaluated according to the score shown in Supplementary Fig. 10B.

**RNA expression profiling by RNA-Seq.** Total RNA from organoids or adenomas was used for RNA-Seq. Random primed complementary DNA libraries were constructed and sequenced using the HiSeq2500 (Illumina) platform by GATC (Konstanz, Germany). Each sample was covered by at least 30 million single reads of 50-bp length.

**Bioinformatics analyses of RNA-Seq and ChIP-Seq data.** RNA-Seq FASTQ files were processed using the RNA-Seq module implemented in the CLC Genomics Workbench v8.0 software (Qiagen Bioinformatics) with default settings and were mapped to the GRCm38/mm10 mouse reference genome and its associated gene and transcript annotation (EMSEMBL). RNA-Seq data were filtered to exclude weakly expressed transcripts with less than five mapped exon reads in all samples from the analysis and subjected to upper quartile normalization using the R/ Bioconductor RUVSeq (remove unwanted variation from RNA-Seq data) package as described in Risso et al.[54]. Differential gene expression analysis was performed with edgeR[55,56] or DESeq2[57] after further normalization using the RUVg approach to remove variation between RNA samples resulting from differences in library preparation. GSEA was performed using the GSEA software[58,59]. Lgr5 + or EphB2[high] stem cell gene sets were obtained from refs.[13–15]. The Wnt/β-catenin target gene sets were obtained from Fevr et al.[60]. The Notch target gene set was obtained from Li et al.[61]. Additional gene sets representing genes involved in Wnt/ β-catenin signaling, EMT or c-Myc target genes were obtained from the Molecular Signatures database (MSigDB)[33]. Enrichment of Hallmark gene sets or Kyoto Encyclopedia of Genes and Genomes (KEGG) pathways among differentially regulated genes was analyzed with the Molecular Signatures database program (MSigDB)[33]. Heatmaps were generated with GENE-E (Broad Institute).

ChIP-Seq data of genome-wide Ap4 occupancy in murine B and T cells (GEO accession nos. GSE80669 and GSE58075) and human CRC cells (GEO accession no. GSE46935) were obtained from previously published studies[7,22,23] and were analyzed with the UCSC genome browser[62].

**In silico analysis of human colorectal patient samples.** Normalized RNA expression (RNA-Seq by expectation maximization (RSEM)) data from CRC patient samples were obtained from the publically available TCGA datasets[31] at https://cancergenome.nih.gov/. Association of TCGA patient samples with the different CMS categories was obtained from the Cancer Subtyping Consortium (CRCSC) at www.synapse.org. The CMS subtypes were described in Guinney et al.[32].

**Statistical analysis.** The Graph Pad Prism software was used for statistical analyses. A Student's t-test (unpaired, two-tailed) was used for calculation of significant differences between two groups of samples or mice, with $p < 0.05$ considered significant. Asterisks generally indicate: $*p < 0.05$, $**p < 0.01$ and $***p < 0.001$, n.s. = not significant. For calculation of correlation coefficients, Pearson's or Spearman's correlation analyses were applied. Kaplan–Meier curves were analyzed by log-rank (Mantel–Cox) test.

## Data availability
RNA-Seq data that support the findings of this study have been deposited in Gene Expression Omnibus (GEO) with the accession codes "GSE99434" and "GSE99437". All other data are available from the corresponding author on reasonable request.

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

## Acknowledgements

We are grateful to Marlon Schneider, Hans Clevers and Klaus-Peter Janssen for providing mice and plasmids, and to Andrea Sendelhofert and Sabine Schäfer for help with IHC and EM detections, respectively. This study was supported by grants of the DFG/ Deutsche Forschungsgemeinschaft (grant He 2701/7-1 and He 2701/7-2) and the Rudolf-Bartling-Stiftung to H.H.

## Author contributions

S.J.: performed the analyses of mice, prepared figures and assisted with manuscript writing; M.K.: performed all bioinformatics analyses and prepared related figures, manuscript editing; R.J.: analysis of human CRC cells; U.G.: supported IHC and analysis/ breeding of mice; S.M.: electron microscopic analysis; S.B.: supported expression analyses of CRC lines and organoid microscopy; D.H.: IHC analysis on human CRC samples; P.J.: provided protocols and supported analyses of organoids/tumoroids, manuscript editing; H.H.: concept and supervision of study, experimental design, wrote manuscript.

## Additional information

**Competing interests:** The authors declare no competing interests.

