## [Peer Review File · Nature Communications]

Reviewers' comments:

Reviewer #1 (Remarks to the Author):

The manuscript by Jaeckel et al: "Ap4 is rate limiting for intestinal tumor formation by controlling the homeostasis of intestinal stem cells" describes the role of transcription factor Ap4 in homeostasis and tumorigenic development in the intestine by generating both a germline and intestinal epithelial cell (IEC) specific knockdown of Ap4 in healthy and Apcmin (familial adenomatous polyposis model) mice. This study was based on observations of co-expression of c-MYC and AP4 in progenitor/transit amplifying cells and a possible role of Ap4 in EMT and colorectal cancer progression was hypothesized.

The paper is written clearly and the data is of very high quality, however, at this point the lack of mechanistic insight prevents this paper from being published in Nat comms. in my opinion. In particular:

- While a role of Ap4 in EMT has not been proven in the used mouse and organoid models, nor an increase in cell proliferation could be observed. However, loss of Ap4 does seem to play a role in the exhaustion of the intestinal stem cell pool in combination with an enrichment in Paneth cells in the small intestine. In contrast, the TCGA cellular data presented in this manuscript does show that high Ap4 expression correlates to c-myc related proliferation and mesenchymal characteristics. These Ap4-high cell lines can be clustered together in the mesenchymal CMS2 subtype of CRC classification. As the in vivo models suggests differentiation of ISCs into Paneth cells in adenomas, whilst the TCGA lines describe a more EMT phenotype in CRC it is unclear how these two observations can be merged into one straightforward conclusion.

- Ap4 expression is present in almost all cells of an Apc^{-/-} adenoma, whilst only present in progenitor cells of the healthy intestine. The question can be raised whether upregulation of the Wnt signaling pathway results in increased Ap4 expression. This is not shown in this manuscript, as it only compares Ap4⁺ and Ap4⁻ adenomas. Furthermore, AP4 has been shown to bind to various Wnt promoters in a mouse CRC cell line which makes Ap4 expression versus Wnt signaling like a "chicken or the egg" paradox. Same holds for Ap4 and the NOTCH1 receptor. This should be elucidated.

- The role of Ap4 in the inhibition of adenoma formation can be described as intestinal epithelial cell-autonomous as the authors show elegantly that Ap4 KD expression patterns found in vivo are maintained in an in vitro organoid setting, thereby excluding microenvironmental regulation. Although the experimental data clearly shows a reduction in ISCs and an increase in Paneth cells, the mechanistic component is lacking. Ap4 IP experiments suggest binding of Ap4 to several Wnt and Notch components, but is unclear which of these factors ultimately leads to Paneth cell differentiation. The authors propose a role for Notch signaling pathway, however, all the experimental data suggests that no conventional Notch pathway inhibition is taking place. Specifically, a general inhibition of Notch by γ -secretase inhibitor DBZ leads to upregulation of the whole secretory lineage instead of solely the Paneth cells.

- In addition, it is also unclear how a decrease in Ap4 expression results in a decrease in the number of stem cells, but not a total depletion of these stem cells. Is the effect of Ap4 important for the maintenance of the adenoma?

- Critically, the experimental data seems to suggest a different function of Ap4 in the colon compared to the small intestine: 1) No differences in adenoma numbers or size could be observed after Ap4 loss in Apcmin mice, and 2) No differences in length and composition of the colon could be observed after Ap4 loss in 'normal' Lgr5 mice.

Is this due to the absence of Paneth cells in the colon? Paneth don't express Ap4 at all. It is unclear how we can translate these findings towards the human setting.

Reviewer #2 (Remarks to the Author):

The manuscript demonstrates convincingly that AP4, the product of the REPIN1 gene, acts downstream of MYC to regulate both the Wnt and Notch pathways in intestinal stem cells and intestinal adenoma stem cells in ApcMin +/-mice. The authors also show that the expression of the proteins encoded by AP4, HES1 and NOTCH1 are correlated in human colorectal cancer samples. The data presented are innovative, the methods are generally rigorous and the presentation is clear. These data advance our understanding of how MYC controls intestinal stem cell fate in normal and neoplastic intestinal epithelia. A number of relatively minor concerns need to be addressed:

1. The demonstration that Notch1 regulates AP4 expression directly is not as clear-cut as other parts of the manuscript. DBZ (or for that matter other gamma secretase inhibitors) have multiple targets. Whether or not endogenous Notch1 regulates AP4 should be demonstrated by knockdown. Although RBPj-k sites exist in the AP4 promoter, it is also possible that the regulation of AP4 by Notch is mediated by c-Myc, a transcriptional target of Notch in other systems. This has to be clarified. Does NICD increase Myc occupancy of the AP4 MYC sites?
2. Which NICD1 antibody was used is unclear. Catalog numbers would help. Cell signaling makes several Notch1 antibodies. Was it the V1774-specific antibody? If so, why was total Notch1 expression not quantified? Along the same lines, Western blots in Figures 4D and S7 lack molecular mass markers, making it difficult to establish band identity
3. While the ApcMin +/- model is an established one, it does have the limitation that it produces small intestinal adenomas rather than colorectal carcinomas (unlike AOM-induced chemical carcinogenesis). This is not a major flaw because the authors do demonstrate expression of AP4 in human colorectal cancers. Nevertheless, this limitation of the experimental system must be acknowledged.

Answers to the Reviewers' comments (in bold):

Reviewer #1 (Remarks to the Author are also provided):

Point 1:

- While a role of Ap4 in EMT has not been proven in the used mouse and organoid models, nor an increase in cell proliferation could be observed. However, loss of Ap4 does seem to play a role in the exhaustion of the intestinal stem cell pool in combination with an enrichment in Paneth cells in the small intestine. In contrast, the TCGA cellular data presented in this manuscript does show that high Ap4 expression correlates to *c-myc* related proliferation and mesenchymal characteristics. These Ap4-high cell lines can be clustered together in the mesenchymal CMS2 subtype of CRC classification. As the *in vivo* models suggests differentiation of ISCs into Paneth cells in adenomas, whilst the TCGA lines describe a more EMT phenotype in CRC it is unclear how these two observations can be merged into one straightforward conclusion.

Answer to point 1:

- **In this point of criticism the reviewer states that the CMS2 subtype represents the mesenchymal subtype of CRCs. However, in the original publication describing the 4 molecular subgroups of CRC (Guinney et al., 2015), which we refer to in our MS, the mesenchymal subtype is CMS4. CMS2 on the other hand represents CRCs with high *c-MYC* expression and *wnt* signaling (with no relation to EMT and/or epithelial vs. mesenchymal characteristics) as we state in the manuscript. Therefore, the argument of the referee stating that we are unable to unite two contradicting observations in our MS is not valid (i.e. the supposedly high AP4 expression in “mesenchymal” CMS2 CRCs versus the enhanced ISC differentiation into Paneth cells in the AP4-KO mouse). Instead we would like to point out, that our analysis of human TCGA data confirms the association of high AP4 expression with elevated *c-MYC* and *wnt* signaling that we also find after NGS analysis of adenomas of *Apc^{Min}* mice with a deletion of *Ap4*. Taken together, we therefore think, that it is not necessary to elaborate further on this point of criticism.**

Point 2:

- Ap4 expression is present in almost all cells of an *Apc*^{-/-} adenoma, whilst only present in progenitor cells of the healthy intestine. The question can be raised whether upregulation of the *Wnt* signaling pathway results in increased Ap4 expression. This is not shown in this manuscript, as it only compares Ap4⁺ and Ap4⁻ adenomas. Furthermore, AP4 has been shown to bind to various *Wnt* promoters in a mouse CRC cell line which makes Ap4 expression versus *Wnt* signaling like a “chicken or the egg” paradox. Same holds for Ap4 and the NOTCH1 receptor. This should be elucidated.

AP4 was previously characterized by us as a *c-MYC* target (Jung and Hermeking, 2009; Jung et al., 2008). In addition, we determined that human CRCs express elevated levels of AP4 (Jung et al., 2008) that correlate with disease progression (Jackstadt et al., 2013). *c-MYC* is a well-known target of the *wnt*/ β -catenin pathway in CRCs and also functionally relevant for CRCs (He et al., 1998; Sansom et al., 2007). Therefore, the general assumption is that AP4 is up-regulated in CRCs, adenomas and in progenitor cells due to the elevated levels of *c-MYC* expression, which is known to be caused by activation of the *wnt*/ β -cat pathway in these scenarios.

To further substantiate these conclusions, we down-regulated the expression of *c-MYC* in DLD-1 cell, which harbor an activated β -catenin allele, by more than 60% using siRNAs (Fig. R1A). This resulted in a significant down-regulation of AP4 expression by more than 60%. Therefore, it can be deduced that the up-regulation of AP4 in CRCs, which generally display an activation of the *wnt*/ β -cat pathway, is largely mediated by elevated *c-MYC* expression. Furthermore, knock-down of β -cat/*ctnnb1* (Fig.R1B,C) also reduced the expression of *c-MYC* and AP4, as well as the *wnt*/ β -cat

target *Axin2*. As suggested by the referee, *AP4* may also be a direct wnt/ β -cat target. To address this issue we performed several additional experiments. However, in these analyses we could not detect any evidence for a direct induction of *AP4* by activated β -catenin: Both, murine CT26 (Fig. R1D) and human HT29 (Fig. R1E) cells did not show induction of *AP4* after activation of ectopic β -cat mutants although other known β -cat/TCF4 targets were induced. In addition, ectopic expression of mutant β -cat was not able to induce an *AP4* promoter fused to luciferase, whereas the so-called TOP/FOP vectors, which are standard wnt- β -cat reporters, were strongly activated in this assay (Fig. R1F). In conclusion, *AP4* is clearly downstream of *c-MYC*, but presumably an indirect target of the wnt/ β -cat pathway. Since deletion of *AP4* attenuated the expression of numerous wnt/ β -cat pathway components in the adenomas of the *Apc^{Min}* mouse model and in untransformed IECs and derived organoids, we conclude that *AP4* contributes to the expression of wnt/ β -cat components. Since *c-MYC* is downstream of wnt/ β -cat signaling, the resulting further increase in *c-MYC* may reinforce the expression of *AP4* and stabilize this regulatory circuit. Therefore, wnt/ β -cat, *c-MYC* and *AP4* presumably form a positive feedback-loop.

Analysis of potential direct regulation of *AP4* by NICD1:

Since we were not able to reproduce the results shown in Supplemental Fig. 7I+J (induction of *AP4* by ectopic expression of NICD1) in Caco2 (Fig. R2A) and also not in DLD-1 (Fig. R2B) or CT26 (Fig. R2C) cells and were not able to detect activation of an *AP4* reporter plasmid after ectopic expression of NICD1 (Fig. R2D), we removed Supplemental Figures 7I-M and the conclusions made from these from the manuscript. Since we now repeated the assays several times using different cell densities and always obtained negative results, the initial detection of an induction of *AP4* was presumably a false positive result. Therefore, the down-regulation of *AP4* after inhibition of the NOTCH pathway using DBZ is presumably a consequence of the down-regulation of *c-MYC*, which represent a known target of NICD1 (now shown as Supplemental Fig. SI,H). The current results indicate that *AP4* is not directly regulated by NICD1, but indirectly via the Notch-mediated regulation of *c-MYC*.

- The role of *Ap4* in the inhibition of adenoma formation can be described as intestinal epithelial cell-autonomous as the authors show elegantly that *Ap4* KD expression patterns found in vivo are maintained in an in vitro organoid setting, thereby excluding microenvironmental regulation. Although the experimental data clearly shows a reduction in ISCs and an increase in Paneth cells, the mechanistic component is lacking. *Ap4* IP experiments suggest binding of *Ap4* to several Wnt and Notch components, but is unclear which of these factors ultimately leads to Paneth cell differentiation. The authors propose a role for Notch signaling pathway, however, all the experimental data suggests that no conventional Notch pathway inhibition is taking place. Specifically, a general inhibition of Notch by γ -secretase inhibitor DBZ leads to upregulation of the whole secretory lineage instead of solely the Paneth cells.

The referee asks for a more detailed mechanism by which deletion of *AP4* promotes the differentiation towards Paneth cells as he states that the observed decrease in Notch pathway activity is not sufficient to explain the observed phenotypes. We agree, that the decrease in Goblet cells combined with the increase in Paneth cells suggests that the effects of *AP4* loss are not solely due to a decrease in the activity of the NOTCH pathway (in that case all secretory cell types would have presumably increased).

In response, we would like to mention that the deletion of *Ap4* did not selectively reduce NOTCH pathway activity, but affected the expression of numerous other factors and pathways. These, in combination with the decreased NOTCH pathway activity, have presumably caused the observed changes in the numbers of specific cell types. Notably, also the wnt pathway is decreased after *Ap4* deletion. It is well known that a controlled balance of the Notch and Wnt pathway activity is important for the homeostasis of stem cells and cell fate decisions in the intestine and that these two pathways regulate each other at multiple points (Tian et al., 2015). Specifically, inhibition of NOTCH1/2 receptors was shown to induce Wnt-signaling, which then promoted goblet cell differentiation. The decrease of wnt-pathway component expression after *Ap4* deletion may therefore alter the expected outcome of a NOTCH inhibition (i.e. increase of all secretory cell numbers) unto the decrease of Goblet cell differentiation observed here. In addition, other factors, that showed deregulation after *Ap4* deletion may have influenced the differentiation of the goblet cells and/or the effects of NOTCH pathway inhibition. For example, *Ap4*-deficiency resulted in a decrease in expression of the transcription factor *Spdef* (results added to Fig. 7B,C). Interestingly, it was previously shown that ectopic expression of *Spdef* in the murine intestine promotes the differentiation of Goblet cells, while it decreased the number of Paneth cells (Noah et al., 2010). Accordingly, the *Ap4*-mediated decrease of *Spdef* could potentially also contribute to the decreased number of Goblet cells detected after *Ap4* deletion. We included these points in the discussion.

As the regulation and analysis of cell fate decision in the intestine is highly complex and still not fully understood we suggest that further functional validation of the role/requirement of single factors that we identified as differentially expressed after *Ap4* deletion should be performed in future studies. In addition, it is likely that a group/combination of factors regulated by *Ap4* is responsible for the observed effects. Therefore, analyzing the sufficiency of single factors in this system bears considerable risk.

- In addition, it is also unclear how a decrease in *Ap4* expression results in a decrease in the number of stem cells, but not a total depletion of these stem cells.

We assume that the partial decrease in NOTCH and Wnt signaling that we observed after deletion of *Ap4* in intestinal epithelial cells was not sufficient to completely deplete intestinal and colonic stem cells. Similar decreases in the number of stem cells were observed after inhibition of the NOTCH pathway. For example, partial inhibition of Notch signaling by deleting *Notch1* (but not *Notch2*) or by treatment with DBZ resulted in a decrease of stem cells (Carulli et al., 2015; VanDussen et al., 2012). However, when Notch signaling was blocked completely, e.g. by simultaneous *NOTCH1* and *NOTCH2* receptors deletion or *RBPJK* deletion, a complete loss of ISCs resulted (Riccio et al., 2008).

Also, complete abrogation of wnt signaling by deletion of *Tcf4* resulted in a complete loss of Lgr5+ ISC (van Es et al., 2012), whereas partial inhibition of Wnt signaling resulted in a reduction of stem cells (Huels et al., 2018).

Is the effect of Ap4 important for the maintenance of the adenoma?

To address the question as to whether Ap4 is important for the maintenance of adenomas we analyzed the effect of acute loss of APC with and without simultaneous deletion of Ap4 in ISC-derived organoids. Therefore, we isolated crypts of the intestine derived from *Lgr5-CreERT2/Apc fl/fl/Ap4 fl/fl* and *Lgr5-CreERT2/Apc fl/fl* mice. 48 hours after plating of crypts in matrigel overlaid with ENR media (contains EGF, Noggin, and RSPO1) we deleted either *Apc* and *Ap4*, or as a control, *Apc* alone in LGR5-positive stem cells of the newly formed intestinal organoids by addition of 4-OHT. After switching culture conditions to EN media devoid of RSPO1 (only APC-deficient tumor organoids can grow under these conditions), we obtained less tumor organoids after *Ap4* deletion when compared to *Ap4*-proficient tumor organoids (see Fig. 4F,G in the revised manuscript). This supports the notion that Ap4 has an important role during tumor-initiation, and it confirms the results we previously obtained *in vivo* using germ-line and IEC-specific deletion of *Ap4*. Importantly, after serial passaging of these tumor organoid cultures, the amount and the size of tumor organoids was not influenced by the deletion of *Ap4* (Fig. 4F,G,H, Supplemental Fig. 4C). Therefore, Ap4 is important for the initiation but not required for the maintenance of tumoroids. These results are in line with the observations we previously made in *Apc^{Min}* mice, where *Ap4* loss decreased the number of adenomas but not their growth/size. These results and their discussion were included in the revised version.

- Critically, the experimental data seems to suggest a different function of Ap4 in the colon compared to the small intestine: 1) No differences in adenoma (in the colon ??) numbers or size could be observed after Ap4 loss in *Apcmin* mice, and 2) No differences in length and composition of the colon could be observed after Ap4 loss in 'normal' *Lgr5* mice ??.

Is this due to the absence of Paneth cells in the colon? Paneth don't express Ap4 at all. It is unclear how we can translate these findings towards the human setting.

In the colon we detected a mean of 1-2 adenomas per mouse (see Supplemental Fig. S2A,I). This quantification was now added to the manuscript. However, due to the low numbers and big difference in numbers of colonic tumors among the mice the differences between mice with different AP4 status were not significant. Nonetheless, there was a trend towards a lower amount of adenomas in colon of *Ap4*-deficient *Apc^{Min}* mice.

As indicated by the referee, the length of colon did not change in mice with *Ap4*-deficiency. However, whether this is due to the absence of Paneth cells in the colon is currently unknown. We would like to mention that so-called deep crypt secretory cells (DCS cells) have been identified in colonic crypts which have been proposed to have Paneth cell like functions. These are positive for Reg4 and are intermingled with CBCs (Sasaki et al., 2016). To determine any potential effects of *Ap4* loss on DCS cells we now performed an analysis of Reg4 expression by immunohistochemistry (Fig. R3A). As expected, the small intestine was negative for Reg4. The amount of Reg4+ DCS cells was not affected by deletion of *Ap4* according to the quantification we performed in 3 mice per genotype (Fig. R3A,B). Therefore, *Ap4* does not seem to regulate the abundance of Reg4-positive cells in the colon. However, since the Reg4-positive cells detected by us were not restricted to the crypt base, we are not completely certain about their classification as Paneth cells. In addition, a recent paper showed that other cells may provide the niche for colonic stem cells (Degirmenci et al., 2018).

As the referee asked for the relevance of our observations for the “human setting” we assumed that he/she asked for parallels between the effects of *AP4* deletion in the small intestine and the colon. As the main effect of *Ap4* loss described here is the decrease of ISCs, we quantified the number of colonic stem cells by *in situ* hybridization of *Smoc2* and *Lgr5* mRNA (Fig. S6I,J in the manuscript). Thereby, we detected that *AP4* loss resulted in a decrease of colonic stem cells by ~50% (Fig. S6I,J in the manuscript). Therefore, the decrease in colonic stem cells is similar to the effect on ISCs. Since we did not detect an increase in *Reg4* cells, which may be the Paneth-like cells of the colon, or any other cells, and only a decrease in the number of Goblet cells, the decrease in the colonic stem cells is presumably not caused by increased differentiation in the absence of *Ap4*, but due to decreased maintenance of the stem cells caused by the deletion of *Ap4*.

We assume that this decrease in the number of colonic stem cells in the absence of *Ap4* would translate into a decreased number of colorectal cancers if studied in a suitable mouse model. Similarly, in humans *AP4* may also have a critical function in the control of colonic stem cell number and thereby influence the incidence of colorectal cancer. These points are now also mentioned in the discussion.

Reviewer #2 (Remarks to the Author):

The manuscript demonstrates convincingly that AP4, the product of the **REPIN1** gene, acts downstream of MYC to regulate both the Wnt and Notch pathways in intestinal stem cells and intestinal adenoma stem cells in ApcMin +/-mice. The authors also show that the expression of the proteins encoded by AP4, HES1 and NOTCH1 are correlated in human colorectal cancer samples. The data presented are innovative, the methods are generally rigorous and the presentation is clear. These data advance our understanding of how MYC controls intestinal stem cell fate in normal and neoplastic intestinal epithelia. A number of relatively minor concerns need to be addressed:

In order to avoid any confusion, we would like to mention that the gene studied/deleted here is *TfAP4* (also known as *AP-4*; *AP4*, *Tcfap4*; *bHLHc41*), which is located on chromosome 16p13.3, *H. sapiens* (Chr. 16, *Mus musculus*), and encodes a B-HLH-LZ transcription factor.

The *REPIN1* gene that was mentioned by the referee, is a different gene located on Chr.7, *H. sapiens* (also known as *AP4*, *RIP60*, *ZNF464*, *Zfp464*) and encodes a Zinc-finger protein that binds to replication origins. In *M. musculus* *Repin1* is located on Chr. 6.

1. The demonstration that Notch1 regulates AP4 expression directly is not as clear-cut as other parts of the manuscript. DBZ (or for that matter other gamma secretase inhibitors) have multiple targets. Whether or not endogenous Notch1 regulates AP4 should be demonstrated by knockdown. Although RBPj-k sites exist in the AP4 promoter, it is also possible that the regulation of AP4 by Notch is mediated by c-Myc, a transcriptional target of Notch in other systems. This has to be clarified. Does NICD increase Myc occupancy of the AP4 MYC sites?

We agree with the referee that this part of the manuscript was not as rigorously performed as may have been necessary to exclude indirect regulation of *AP4* after NOTCH pathway activation or inhibition. Specifically, it did not exclude the possibility of indirect regulation of *AP4* after DBZ treatment (e.g. via c-MYC). While performing additional experiments along these lines we encountered problems in reproducing the induction of *AP4* after activation of an inducible NICD1 allele shown in Figure S7J. In addition, another human CRC cell line (DLD-1) and the murine line CT26 also did not display activation of *AP4* by ectopic NICD1 expression (Fig. R2A-C). Furthermore, a murine Ap4 reporter was not induced by ectopic NICD1 expression (see Fig. R2D), indicating that the regulations shown in Fig. S7I to S7M (of the previous submission) may not be conserved between species. Therefore, we decided to remove Figures S7I to S7M from the revised version of the MS. In the results we now mention that *AP4* may be indirectly regulated via the decrease of c-MYC that is known to occur after treatment with DBZ (since c-MYC is a known target of the NOTCH pathway (Palomero et al., 2006)). The decrease of c-MYC expression after NOTCH pathway inhibition was also detected in the current Figures S8H and S8I. However, since this potential feedback regulation between *AP4* and c-MYC is not the focus of the study and should be addressed in more detail in a separate study, we decided to not further address this issue in the manuscript.

2. Which NICD1 antibody was used is unclear. Catalog numbers would help. Cell signaling makes several Notch1 antibodies. Was it the V1774-specific antibody? If so, why was total Notch1 expression not quantified? Along the same lines, Western blots in Figures 4D and S7 lack molecular mass markers, making it difficult to establish band identity

The antibody used for detection of NICD1 was indeed the cleaved Notch1 (Val 1744) (D3B8) Rabbit mAB #4147 antibody from Cell Signaling.

In publications describing the analysis of NOTCH pathway activity the detection of the cleaved Notch1 (NICD1) without simultaneous detection of the full length NOTCH1 protein is commonly used for determination of the activity of the NOTCH pathway. (For example: (Sarkar et al., 2017; Srinivasan et al., 2016; Tian et al., 2015; Zhou et al., 2011)). As the detection of the NOTCH1 precursor would not add necessary information to the manuscript, we did not do it.

3. While the *Apc*^{Min} +/- model is an established one, it does have the limitation that it produces small intestinal adenomas rather than colorectal carcinomas (unlike AOM-induced chemical carcinogenesis). This is not a major flaw because the authors do demonstrate expression of AP4 in human colorectal cancers. Nevertheless, this limitation of the experimental system must be acknowledged.

The limitation of the *Apc*^{Min} mouse model is now mentioned accordingly in the discussion:

The *Apc*^{Min} mouse model has the limitation that most tumors are located in the small intestine rather than in the colon. Since many mechanisms and pathways are conserved between intestinal and colonic tumors, this model may also provide insights into the biology of human CRCs and FAP, the inherited syndrome associated with germ-line *APC* mutations. Since we provide evidence that the up-regulation of *AP4* in human CRC samples is associated with activation of the same pathways that we identified in the mouse model, the findings presented here are likely to play a role in human CRC.

Figure R1

Ap4-is not a direct target of CTTNB1

(A) DLD-1 colorectal cancer cells were transfected with a *CTNNB1*-specific siRNA the *Lipofectamine RNAiMAX* Transfection Reagent (Thermo Fisher Scientific) according to manufacturer's instructions at a concentration of 10 nM. 48 h later, RNA were subjected to q-PCR analysis of the indicated mRNAs. (B) DLD-1 colorectal cancer cells were transfected with a *CTNNB1*-specific siRNA with the

Lipofectamine RNAiMAX Transfection Reagent (Thermo Fisher Scientific) according to manufacturer's instructions at a concentration of 10 nM. 48 h later, RNA were subjected to q-PCR analysis of the indicated mRNAs. (C) SW480 colorectal cancer cells were transfected with a *CTNNB1*-specific siRNA the *Lipofectamine RNAiMAX* Transfection Reagent (Thermo Fisher Scientific) according to manufacturer's instructions at a concentration of 10 nM. 48 h later, protein lysates were subjected to q-PCR analysis of the indicated proteins. (D) CT-26 cells were stably transduced with either *CTNNB1_S33Y* encoding or empty (CTRL) lentiviral particles. 72 hrs after treatment of cells with doxycycline for induction of S33Y, gene expression for the Wnt target gene *Axin2* and for *Ap4* was analyzed by qRT-PCR. Fold changes are in relation to non-DOX treated cells, and the house-keeper *B2m* was used for normalization. The experiment was performed in biological triplicates. Error bars indicate standard deviation. Significance of fold changes was calculated using Multiple T-tests. *CTNNB1-S33Y* = refers to the mutant human *CTNNB1* version harboring the activating S33Y mutation. It's expression can only be detected in the S33Y virus-transduced murine CT26 cells. (E) HT29 colorectal cancer cells were stably transduced with either *CTNNB1_S33Y* (S33Y) encoding or control lentiviral particles. 72 hrs after treatment of cells with doxycycline, gene expression for bona-fide Wnt target genes *NKD1*, *AXIN2*, and *LGR5* and for *AP4* were analyzed by qRT-PCR in cells expressing *CTNNB1_S33Y* (S33Y DOX) or controls (CTRL DOX). CTRL DOX I was set to „1“, and the house keeper genes *B2M* and *PPIA* were used for normalization. The experiment was performed in biological duplicates (I and II). Error bars indicate technical standard error of the mean (SEM). Multiple t-tests have been applied to calculate the significance of expression differences between the biological replicates DOX I+II versus CTRL I+II. *CTNNB1*^o: Primers used for qRT-PCR cannot distinguish between endogenous wild-type and ectopic S33Y mutant *CTNNB1*. Hence, the data above show that treatment of cells with doxycycline for 72 hrs led to a mutant *CTNNB1-S33Y* level which was 4-5 times higher than the level of endogenous *CTNNB1* in HT29 cells. (F) Luciferase Reporter Assay: a 964bp and 1085bp fragment of the murine *Tfap4* promoter harboring the transcription start sites of two different annotated *Tfap4* mRNA isoforms (NM_031182.2 and ENSMUST00000005862.8, respectively) were PCR-amplified from genomic DNA from SV/C57Bl6 mice and cloned into pBV-Luc via NheI and HindIII. All constructs were verified by Sanger sequencing. CT26 were seeded at 3×10^4 cells per 12-well and transfected the next day with 100ng reporter plasmid, 100ng effector plasmid and 10ng Renilla plasmid for normalization. Cells were lysed after 48h and subjected to dual reporter assays (Promega) according to manufacturer's instructions. Luminescence intensities were measured with an Orion II luminometer (Berthold) in 96-well format and analyzed with the SIMPLICITY software package (DLR). C,D,E,F: Results represent the mean +/- SD. Results were subjected to an unpaired, two tailed Student's *t*-test with p-values * < 0.05, ** < 0.01, *** < 0.001, n.s. = not significant.

Figure R2

Ap4 is not a direct target of Notch1/NICD1

(A) qPCR analysis of the indicated mRNAs after ectopic NICD1 expression from a pRTR-NICD1 episomal plasmid in CaCo2 CRC cells exposed to DOX for the indicated periods. Results represent the mean \pm SD (n=3). (B) qPCR analysis of the indicated mRNAs after ectopic NICD1 expression from a pRTR-NICD1 episomal plasmid in DLD-1 CRC cells exposed to DOX for the indicated periods. Results represent the mean \pm SD (n=3). (C) CT26 cells were transfected at 2×10^5 cells / 6-well and transfected the next day with 4 μ g pcDNA3.1 or pcDNA-mNICD, respectively. Cells were harvested after 60h and subjected to RNA extraction and cDNA synthesis. QPCR calculation as described : Fold changes normalized to control transfected cells (pcDNA) and B2m. (D) Luciferase Reporter Assay: a 964bp and 1085bp fragment of the murine *Ap4* promoter harboring the transcription start sites of two different annotated *Ap4* mRNA isoforms (NM_031182.2 and ENSMUST00000005862.8, respectively) were PCR-amplified from genomic DNA from SV/C57Bl6 mice and cloned into pBV-Luc via NheI and HindIII. All constructs were verified by Sanger sequencing. CT26 were seeded at 3×10^4 cells per 12-well and transfected the next day with 100ng reporter plasmid, 100ng effector plasmid and 10ng Renilla plasmid for normalization. Cells were lysed after 48h and subjected to dual reporter assays (Promega) according to manufacturer's instructions. Luminescence intensities were measured with an Orion II luminometer (Berthold) in 96-well format and analyzed with the SIMPLICITY software package (DLR). A,B,C,D: Results represent the mean \pm SD. Results were subjected to an unpaired, two tailed Student's *t*-test with p-values * < 0.05, ** < 0.01, *** < 0.001, n.s. = not significant.

Figure R3

(A) Immunohistochemical detection of Reg4 (R&D Systems) in colon from mice of the indicated genotype. Counterstaining with hematoxylin. Scale bar = 25 μ m. (B) 180 crypts from 1 male and 2 female mice were analyzed for Reg4 positive cells per genotype. Results represent the mean \pm SD. Results were subjected to an unpaired, two tailed Student's *t*-test with *p*-values * < 0.05, ** < 0.01, *** < 0.001, n.s. = not significant.

Rebuttal references

Carulli, A.J., Keeley, T.M., Demitrack, E.S., Chung, J., Maillard, I., and Samuelson, L.C. (2015). Notch receptor regulation of intestinal stem cell homeostasis and crypt regeneration. *Dev Biol* 402, 98-108.

Degirmenci, B., Valenta, T., Dimitrieva, S., Hausmann, G., and Basler, K. (2018). GLI1-expressing mesenchymal cells form the essential Wnt-secreting niche for colon stem cells. *Nature* 558, 449-453.

Guinney, J., Dienstmann, R., Wang, X., de Reynies, A., Schlicker, A., Soneson, C., Marisa, L., Roepman, P., Nyamundanda, G., Angelino, P., *et al.* (2015). The consensus molecular subtypes of colorectal cancer. *Nat Med* 21, 1350-1356.

He, T.C., Sparks, A.B., Rago, C., Hermeking, H., Zawel, L., da Costa, L.T., Morin, P.J., Vogelstein, B., and Kinzler, K.W. (1998). Identification of c-MYC as a target of the APC pathway. *Science* 281, 1509-1512.

Huels, D.J., Bruens, L., Hodder, M.C., Cammareri, P., Campbell, A.D., Ridgway, R.A., Gay, D.M., Solar-Abboud, M., Faller, W.J., Nixon, C., *et al.* (2018). Wnt ligands influence tumour initiation by controlling the number of intestinal stem cells. *Nat Commun* 9, 1132.

Jackstadt, R., Roh, S., Neumann, J., Jung, P., Hoffmann, R., Horst, D., Berens, C., Bornkamm, G.W., Kirchner, T., Menssen, A., *et al.* (2013). AP4 is a mediator of epithelial-mesenchymal transition and metastasis in colorectal cancer. *J Exp Med* 210, 1331-1350.

Jung, P., and Hermeking, H. (2009). The c-MYC-AP4-p21 cascade. *Cell Cycle* 8, 982-989.

Jung, P., Menssen, A., Mayr, D., and Hermeking, H. (2008). AP4 encodes a c-MYC-inducible repressor of p21. *Proc Natl Acad Sci U S A* 105, 15046-15051.

Noah, T.K., Kazanjian, A., Whitsett, J., and Shroyer, N.F. (2010). SAM pointed domain ETS factor (SPDEF) regulates terminal differentiation and maturation of intestinal goblet cells. *Experimental Cell Research* 316, 452-465.

Palomero, T., Lim, W.K., Odom, D.T., Sulis, M.L., Real, P.J., Margolin, A., Barnes, K.C., O'Neil, J., Neuberg, D., Weng, A.P., *et al.* (2006). NOTCH1 directly regulates c-MYC and activates a feed-forward-loop transcriptional network promoting leukemic cell growth. *Proc Natl Acad Sci U S A* 103, 18261-18266.

Riccio, O., van Gijn, M.E., Bezdek, A.C., Pellegrinet, L., van Es, J.H., Zimmer-Strobl, U., Strobl, L.J., Honjo, T., Clevers, H., and Radtke, F. (2008). Loss of intestinal crypt progenitor cells owing to inactivation of both Notch1 and Notch2 is accompanied by derepression of CDK inhibitors p27Kip1 and p57Kip2. *EMBO Rep* 9, 377-383.

Sansom, O.J., Meniel, V.S., Muncan, V., Phesse, T.J., Wilkins, J.A., Reed, K.R., Vass, J.K., Athineos, D., Clevers, H., and Clarke, A.R. (2007). Myc deletion rescues Apc deficiency in the small intestine. *Nature* 446, 676-679.

Sarkar, S., Mirzaei, R., Zemp, F.J., Wei, W., Senger, D.L., Robbins, S.M., and Yong, V.W. (2017). Activation of NOTCH Signaling by Tenascin-C Promotes Growth of Human Brain Tumor-Initiating Cells. *Cancer Res* 77, 3231-3243.

Sasaki, N., Sachs, N., Wiebrands, K., Ellenbroek, S.I., Fumagalli, A., Lyubimova, A., Begthel, H., van den Born, M., van Es, J.H., Karthaus, W.R., *et al.* (2016). Reg4+ deep crypt secretory cells function as epithelial niche for Lgr5+ stem cells in colon. *Proc Natl Acad Sci U S A* 113, E5399-5407.

Srinivasan, T., Than, E.B., Bu, P., Tung, K.L., Chen, K.Y., Augenlicht, L., Lipkin, S.M., and Shen, X. (2016). Notch signalling regulates asymmetric division and inter-conversion between Lgr5 and bmi1 expressing intestinal stem cells. *Sci Rep* 6, 26069.

Tian, H., Biehs, B., Chiu, C., Siebel, C.W., Wu, Y., Costa, M., de Sauvage, F.J., and Klein, O.D. (2015). Opposing activities of notch and wnt signaling regulate intestinal stem cells and gut homeostasis. *Cell Rep* 11, 33-42.

van Es, J.H., Haegebarth, A., Kujala, P., Itzkovitz, S., Koo, B.K., Boj, S.F., Korving, J., van den Born, M., van Oudenaarden, A., Robine, S., *et al.* (2012). A critical role for the Wnt effector Tcf4 in adult intestinal homeostatic self-renewal. *Mol Cell Biol* 32, 1918-1927.

VanDussen, K.L., Carulli, A.J., Keeley, T.M., Patel, S.R., Puthoff, B.J., Magness, S.T., Tran, I.T., Maillard, I., Siebel, C., Kolterud, A., *et al.* (2012). Notch signaling modulates proliferation and differentiation of intestinal crypt base columnar stem cells. *Development* 139, 488-497.

Zhou, D., Zhang, Y., Wu, H., Barry, E., Yin, Y., Lawrence, E., Dawson, D., Willis, J.E., Markowitz, S.D., Camargo, F.D., *et al.* (2011). Mst1 and Mst2 protein kinases restrain intestinal stem cell proliferation

and colonic tumorigenesis by inhibition of Yes-associated protein (Yap) overabundance. Proc Natl Acad Sci U S A *108*, E1312-1320.

REVIEWERS' COMMENTS:

Reviewer #1 (Remarks to the Author):

The rebuttal by Jaeckel et al. "Ap4 is rate limiting for intestinal tumor formation by controlling the homeostasis of intestinal stem cells" addresses all concerns there were after review of the first version. In my opinion this manuscript is eligible for publication in Nature Communications.

This manuscript validates once more that intestinal stem cells (ISCs) are the cells of origin driving adenoma formation, and that reducing ISC numbers can greatly diminish the chances of developing adenoma's. Ap4 is an interesting target that functions in both Notch and Wnt pathways, where it can influence stem/differentiation cell states but leaves cell proliferation unaffected. Further research should be performed to gain more mechanistic insight in Ap4 and its interactions. Especially, it would be interesting to see if inhibition/KO of Ap4 could prevent metastasis.

Reviewer #2 (Remarks to the Author):

The authors address my main concern, and now show -as I had suspected- that the regulation of TFAP4 by NOTCH1 is not direct but most likely mediated by MYC. I still would have liked a quantification of total Notch1 protein, but I don't consider this a major concern.